# Context Matters: Leveraging Contextual Features for Time Series Forecasting

## Abstract

Time series forecasts are often influenced by exogenous contextual features in addition to their corresponding history. For example, in financial settings, it is hard to accurately predict a stock price without considering public sentiments and policy decisions in the form of news articles, tweets, etc. Though this is common knowledge, the current state-of-the-art (SOTA) forecasting models fail to incorporate such contextual information, owing to its heterogeneity and multimodal nature. To address this, we introduce ContextFormer, a novel plug-and-play method to surgically integrate multimodal contextual information into existing pre-trained forecasting models. ContextFormer effectively distills forecast-specific information from rich multimodal contexts, including categorical, continuous, time-varying, and even textual information, to significantly enhance the performance of existing base forecasters. ContextFormer outperforms SOTA forecasting models by up to 30% on a range of real-world datasets spanning energy, traffic, environmental, and financial domains.

## 1 Introduction

Numerous state-of-the-art (SOTA) solutions to time series forecasting (Lin et al., 2021) have predominantly depended only on the time series history. However, in many real-world forecasting applications, such as predicting stock prices, air quality, or household energy consumption, future values are frequently influenced by external contextual factors like geographical and economic indicators. Industrial solutions for forecasting, such as predicting the demand for online food delivery (Chad Akkoyun, 2022), have shown the potential to improve forecasting accuracy by incorporating macroeconomic factors like tax refunds.

However, the current SOTA forecasting models (Liu et al., 2024; Nie et al., 2023) still are unable to handle these contextual factors and solely rely on the historical time series to predict the future. We attribute this to the inherent diversity and multimodality of these contextual factors. For example, consider the task of predicting the price of a stock. The contextual factors can vary from categorical indicators like stock category (e.g., energy, technology, or healthcare), continuous and time-varying indicators like market cap and interest rates, or even textual information in the form of news articles. We refer to this multimodal contextual information as metadata and use both these terms interchangeably in this work. Incorporating metadata into forecasting models is hard for the following reasons:

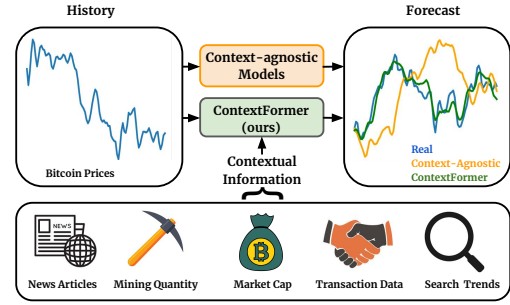

Figure 1: **Forecasting with context.** A context-aware forecaster like ContextFormer can incorporate multimodal contextual information, such as daily news articles, online search trends, and market data, to enhance the accuracy of time-series forecasts.

1. **Lack of multimodal metadata encoders.** We note that the time series domain lacks the availability of foundation models trained on multimodal datasets to extract aligned representations (e.g., CLIP Radford et al. (2021)). These are key to mapping the time series history and multimodal metadata into the same representation space from which the forecast can be decoded.

2. **Non-uniformity in metadata across datasets.** The metadata associated with stock price prediction, such as new articles, tweets and opinions, interest rates, etc., are completely different from the metadata associated with weather prediction, such as rainfall levels, pollution levels, wind direction, and speed, etc. This prevents us from pooling datasets together to train a *context-aware* foundation model for forecasting.

3. **Diversity of metadata within datasets.** For a given dataset, the metadata could be categorical (e.g., national holidays), continuous (e.g., interest rates), or even time-varying (e.g., oil prices). Current approaches often end up modeling such diverse metadata through simple linear regressors (Das et al., 2024), which may be insufficient to capture the complex correlations.

Consequently, for these exact reasons, we note that the recent wave of foundation models for forecasting relies only on history. To this end, we propose a plug-and-play approach to build *context-aware* forecasting models on top of *context-agnostic* SOTA forecasting models. Our approach includes novel architectural additions to handle categorical, continuous, time-varying, and even textual metadata. Additionally, we introduce novel training modifications to ensure that the *context-aware* forecast is at least as good as the *context-agnostic* forecast with respect to the traditional forecasting metrics. Our architectural and training modifications are inspired by theoretical insights on improving any regression model with new, correlated features. Our primary contributions are as follows:

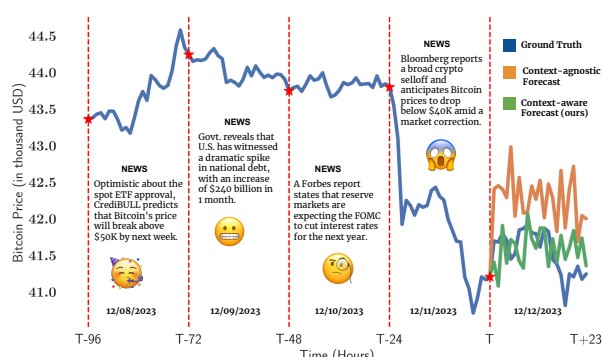

Figure 2: **ContextFormer estimates the true effect of the explainable contextual factors on the forecast**. Here, we show an example of a bitcoin price forecast using news articles and the historical price for the past four days. Note that the sharp decline in price is attributed to an ongoing market correction. Existing *context-agnostic* forecasting models treat this as a transient shock, leading to an overcorrection in price recovery. In contrast, our *ContextFormer* comprehends the underlying market dynamics, resulting in a more accurate and reliable forecast.

1. We propose **ContextFormer**, a novel framework for incorporating diverse multimodal metadata into any context-agnostic forecasting model. **ContextFormer** surgically inserts aligned representation of metadata using cross-attention blocks (Vaswani et al., 2017) into the existing forecasting model architectures.

2. We introduce a plug-and-play fine-tuning approach to effectively incorporate metadata and ensure that the resulting forecasting performance is at least as good as that of the context-agnostic base model.

3. We show definitive improvements in the forecasting performance of state-of-the-art context-agnostic forecasting models, such as PatchTST (Nie et al., 2023) and iTransformer (Liu et al., 2024) across a wide range of real-world forecasting tasks spanning retail, finance, energy, and environmental domains.

## 2 RELATED WORKS

**Classical methods.** These methods predict future values for time series using statistical techniques. Established approaches include ARMA (AutoRegressive Moving Average), which captures temporal dependencies through autoregression and moving averages, and exponential smoothing methods like Holt-Winters and STL (Seasonal-trend decomposition using LOESS), which account for trends and seasonality. The Box-Jenkins methodology is also used for building models to handle non-stationary data (Shumway & Stoffer, 2017; Hyndman & Athanasopoulos, 2018). These techniques have long been the foundation of time series forecasting, providing reliable ways to analyze past trends and make accurate predictions. Prophet (Taylor & Letham, 2017), developed by Facebook, builds upon the traditional techniques by incorporating additional features such as holiday effects and non-linear trends, thus providing greater flexibility and accuracy. Prophet is capable of manag-

ing complex seasonal patterns and irregularities and is known for its robustness in handling missing data and outliers. Classical time series forecasters, despite their capability to integrate covariates, often struggle against deep learning models because they lack the adaptability to fully utilize large datasets and automatically learn intricate temporal dependencies.

**Deep-learning methods.** These methods have been the go-to method to learn the time series features perform forecasting, utilizing the recent advancements in neural network architectures, such as RNNs (Sherstinsky, 2020) and transformers (Vaswani et al., 2017). Notable RNN-based approaches include DeepAR (Salinas et al., 2019) and LSTNet (Lai et al., 2018). SOTA transformer-based approaches include iTransformer (Liu et al., 2024), which applies the attention and feed-forward network on the inverted dimensions and PatchTST (Nie et al., 2023), a channel-independent transformer which takes time series segmented into subseries-level patches as input tokens. Other prominent approaches include Autoformer (Wu et al., 2021), Informer (Zhou et al., 2021), FEDformer (Zhou et al., 2022) and TimesNet (Wu et al., 2022). Recent works have focused on foundation models, which are deep models pre-trained on large amounts of data, enabling them to learn extensive information and a variety of patterns. They can be fine-tuned or adapted to specific tasks with relatively small amounts of task-specific data, showcasing remarkable flexibility and efficiency. Popular foundation models include Time-LLM (Jin et al., 2024), Chronos (Ansari et al., 2024), Lag-Llama (Rasul et al., 2024), and TimesFM (Das et al., 2024). Existing deep learning forecasters fail against context-aware models because they lack the ability to incorporate external factors and dynamic contextual information into predictions.

**Forecasting with covariates.** One of the earliest approaches to conditional forecasting was proposed by Borovykh et al. (2018), which employed a CNN-based model with dilated convolutions to capture extensive historical data for improved forecasting. Recent advancements include models such as TFT (Lim et al., 2021), NBEATSx (Olivares et al., 2023), TiDE (Das et al., 2023), TSMixer (Chen et al., 2023), and TimeXer (Wang et al., 2024b), which integrate covariates in various ways. For instance, TiDE uses dense MLPs to encode past time series data and decode it with future covariates, while TimeXer employs transformer-based architectures to incorporate metadata. LLM-based models like Ploutos (Tong et al., 2024) embed metadata as part of textual queries. Some pre-trained models, such as TimesFM (Das et al., 2024), have extended fine-tuning capabilities to handle covariates through exogenous linear models (see Appendix B), though they require access to future covariate values. Among the other time-series models, TimeWeaver (Narasimhan et al., 2024), a diffusion-based framework for conditional synthesis, integrates metadata using attention-based encoders. Acknowledging the results shown by the aforementioned models, we propose a novel approach to build such context-aware forecasting models from the existing context-agnostic architectures. Our technique enables these models to effectively incorporate contextual information while preserving and utilizing the time-series features acquired by the base models during pre-training.

## 3 PROBLEM FORMULATION

In this section, we formally define the context-aware time series forecasting problem. We are given a multivariate time series $\boldsymbol{X}_{\text{hist}} = \left(\boldsymbol{x}^1, \boldsymbol{x}^2, \ldots, \boldsymbol{x}^L\right)$, with $\boldsymbol{x}^i \in \mathbb{R}^F$. Here, $L$ denotes the history time series length, while $F$ represents the number of channels. Each sample $\boldsymbol{x}^i$ is associated with contextual metadata $\boldsymbol{c}^i$, comprising both categorical features $\boldsymbol{c}_{\text{cat}}^i \in \mathbb{N}^{K_{\text{cat}}}$ and continuous features $\boldsymbol{c}_{\text{cont}}^i \in \mathbb{R}^{K_{\text{cont}}}$. The symbols $K_{\text{cat}}$ and $K_{\text{cont}}$ represent the number of categorical and continuous metadata features, respectively. Together, these features form a vector $\boldsymbol{c}^i = \boldsymbol{c}_{\text{cat}}^i \oplus \boldsymbol{c}_{\text{cont}}^i$, where $\oplus$ denotes vector concatenation. Note that $\boldsymbol{c}^i$ can include both time-varying and time-invariant metadata features. Now, we can define a metadata sequence as $\boldsymbol{C}_{\text{hist}} = \left(\boldsymbol{c}^1, \boldsymbol{c}^2, \ldots, \boldsymbol{c}^L\right)$, having the same number of timesteps as $\boldsymbol{X}_{\text{hist}}$.

To understand this better, let us take the example of the Beijing AQ dataset, which contains the time series data of six air pollutant concentrations: $CO$, $NO_2$, $SO_2$, $O_3$, $PM2.5$, and $PM10$ concentration ($F = 6$), sampled on an hourly basis for four days ($L = 96$). The metadata here includes information on the location, air pressure, amount of rainfall, temperature, dew point, wind speed, and wind direction for each time series sample. In this dataset, location (12 unique labels) and wind direction (17 unique labels) are categorical values ($K_{\text{cat}} = 2$), while the other features are continuous ($K_{\text{cont}} = 5$). All metadata except for location is time-varying. Therefore, given such historical time series data and its paired metadata, the task is to predict the future data samples $\boldsymbol{X}_{\text{future}} = \left(\boldsymbol{x}^{L+1}, \boldsymbol{x}^{L+2}, \ldots, \boldsymbol{x}^{L+T}\right)$ for a forecasting horizon $T$.

We now go on to define a dataset $\mathcal{D} = \{(\boldsymbol{X}_{\text{hist}}^n, \boldsymbol{C}_{\text{hist}}^n, \boldsymbol{X}_{\text{future}}^n)\}_{n=1}^{N}$, which consists of $N$ independent and identically distributed $(\boldsymbol{X}_{\text{hist}}^n, \boldsymbol{C}_{\text{hist}}^n, \boldsymbol{X}_{\text{future}}^n)$ triplets sampled from a joint distribution $P_{\text{data}}$. Formally, the context-aware time series forecasting problem is stated as follows.

**Problem.** For a dataset $\mathcal{D}$ with history length $L$ and forecasting horizon $T$, the goal is to learn the parameters of a model $f(\boldsymbol{X}_{\text{hist}}, \boldsymbol{C}_{\text{hist}}; \theta_{\text{forecast}})$ that predicts the forecast $\hat{\boldsymbol{X}}_{\text{future}} = (\hat{\boldsymbol{x}}^{L+1}, \hat{\boldsymbol{x}}^{L+2}, \ldots, \hat{\boldsymbol{x}}^{L+T})$ for the input $(\boldsymbol{X}_{\text{hist}}, \boldsymbol{C}_{\text{hist}})$, where $(\boldsymbol{X}_{\text{hist}}, \boldsymbol{C}_{\text{hist}}, \boldsymbol{X}_{\text{future}}) \in \mathcal{D}$, such that the following loss function is minimized.

$$\mathcal{L}(\theta_{\text{forecast}}) = \mathbb{E}_{\boldsymbol{X}_{\text{hist}}, \boldsymbol{C}_{\text{hist}}, \boldsymbol{x}_{\text{future}} \sim P_{\text{data}}} \|\boldsymbol{X}_{\text{future}} - \hat{\boldsymbol{X}}_{\text{future}}\|_2, \tag{1}$$

$$\text{where } \hat{\boldsymbol{X}}_{\text{future}} = f(\boldsymbol{X}_{\text{hist}}, \boldsymbol{C}_{\text{hist}}; \theta_{\text{forecast}}). \tag{2}$$

Note that here $\|.\|_2$ represents the $l_2$ norm, while $\mathbb{E}$ denotes the expectation over a distribution. Thus, the optimal learned parameters are given by

$$\theta_{\text{forecast}}^* = \arg\min_{\theta_{\text{forecast}}} \mathcal{L}(\theta_{\text{forecast}}). \tag{3}$$

The above forecasting model is said to be context-aware as the forecast $\hat{\boldsymbol{X}}_{\text{future}}$ is modeled on both $\boldsymbol{X}_{\text{hist}}$ and $\boldsymbol{C}_{\text{hist}}$. In contrast, for a *context-agnostic* model, the forecast will be based only on the history time series $\boldsymbol{X}_{\text{hist}}$, with no contribution from the contextual metadata.

## 4 THEORETICAL MOTIVATION

This section provides theoretical justifications for enhancing forecasting accuracy by incorporating context. From an information-theoretic perspective, we show that including context reduces forecasting uncertainty, thereby improving model accuracy. We then examine the integration of contextual information into a simple linear regression model, illustrating how this approach improves the performance of a simple autoregressive forecaster.

### 4.1 A PERSPECTIVE FROM INFORMATION THEORY

Taking inspiration from a recent work on retrieval-based forecasting (Jing et al. (2022)), we illustrate the relationship between the variables $\boldsymbol{X}_{\text{hist}}$, $\boldsymbol{C}_{\text{hist}}$, $\hat{\boldsymbol{X}}_{\text{future}}$, and $\boldsymbol{X}_{\text{future}}$ for context-aware and context-agnostic forecasters in the form of the graphical models given in Fig. 3. On analyzing the models from an information theoretic perspective, we can show that

$$\mathcal{I}(\boldsymbol{X}_{\text{future}}; \boldsymbol{X}_{\text{hist}}, \boldsymbol{C}_{\text{hist}}) \geq \mathcal{I}(\boldsymbol{X}_{\text{future}}; \boldsymbol{X}_{\text{hist}}) \tag{4}$$

where the quantity $\mathcal{I}(A; B)$ represents the mutual information between the variables $A$ and $B$. This inequality stems from the fact that $\mathcal{I}(\boldsymbol{X}_{\text{future}}; \boldsymbol{X}_{\text{hist}}, \boldsymbol{C}_{\text{hist}}) = \mathcal{I}(\boldsymbol{X}_{\text{future}}; \boldsymbol{X}_{\text{hist}}) + \mathcal{I}(\boldsymbol{X}_{\text{future}}; \boldsymbol{C}_{\text{hist}}|\boldsymbol{X}_{\text{hist}})$ and $\mathcal{I}(\boldsymbol{X}_{\text{future}}; \boldsymbol{C}_{\text{hist}}|\boldsymbol{X}_{\text{hist}}) \geq 0$ for any value of $(\boldsymbol{X}_{\text{future}}, \boldsymbol{X}_{\text{hist}}, \boldsymbol{C}_{\text{hist}})$.

The increase in mutual information between the input variables and the forecast, driven by the inclusion of contextual information, demonstrates that adding any relevant metadata always reduces forecast uncertainty. Additionally, under the premise of commonly assumed Gaussian noise between $\boldsymbol{X}_{\text{future}}$ and $\hat{\boldsymbol{X}}_{\text{future}}$, maximizing mutual information inherently corresponds to minimizing the MSE loss (Jing et al., 2022). This statement can be mathematically expressed as

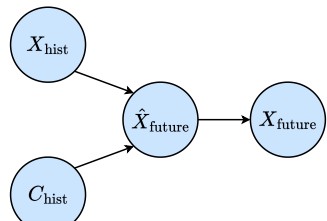

(a) Context-aware Forecaster

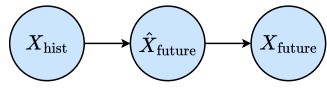

(b) Context-agnostic Forecaster

Figure 3: **Graphical models for forecasting.** The figures represent the graphical models for the two forecasting approaches. In the context-aware model, the forecast $\hat{\boldsymbol{X}}_{\text{future}}$ follows from both the history $\boldsymbol{X}_{\text{hist}}$ and context $\boldsymbol{C}_{\text{hist}}$, while for the context-agnostic model, $\hat{\boldsymbol{X}}_{\text{future}}$ depends only on $\boldsymbol{X}_{\text{hist}}$.

$$\min \mathbb{E}_{P_{\text{data}}} \|\boldsymbol{X}_{\text{future}} - \hat{\boldsymbol{X}}_{\text{future}}\|_2 \iff \max \mathcal{I}\left(\boldsymbol{X}_{\text{future}}; \hat{\boldsymbol{X}}_{\text{future}}\right), \tag{5}$$

thus, context-aware models (Fig. 3a) are more suitable for forecasting under an MSE loss objective. Further discussion on Eq. 5 has been provided in Appendix A. Having demonstrated how incorporating contextual information can enhance the forecasting accuracy of general forecasting models, we now turn our attention to integrating context into a simple autoregressive model. We will also explore the guarantees we can provide for this approach.

## 4.2 Adding Context to an Autoregressive Forecaster

Let $\boldsymbol{y}^t$ be a time-varying quantity influenced by past values and additional contextual information. Here, the underlying assumption is that the true forecast $\boldsymbol{y}^t$ is a linear combination of $p$ lag terms and $q$ context terms. First, we model $\boldsymbol{y}^t$ only as a $p$-order autoregressive (AR) process $\boldsymbol{y}^t = \boldsymbol{x}^t \boldsymbol{\beta} + \boldsymbol{\epsilon}$. Here, $\boldsymbol{x}^t = \begin{bmatrix} \boldsymbol{y}^{t-1} & \boldsymbol{y}^{t-2} & \cdots & \boldsymbol{y}^{t-p} \end{bmatrix}$ is a vector of the previous $p$ lagged values, $\boldsymbol{\beta}$ is a vector of AR coefficients with $\boldsymbol{\beta} \in \mathbb{R}^p$, and $\boldsymbol{\epsilon}$ is the error term. Note that even though $\boldsymbol{y}_t$ depends on additional contextual information, the assumed linear model only depends on the lag parameters, reflecting the context-agnostic case.

Given $n + p$ observations, we construct an $n \times p$ matrix $\boldsymbol{X}$ of lagged values, allowing the model to be written as $\boldsymbol{Y} = \boldsymbol{X}\boldsymbol{\beta} + \boldsymbol{\epsilon}$, where $\boldsymbol{Y}$ is the vector of observed values and $\boldsymbol{Y} \in \mathbb{R}^n$. This formulation enables the least squares estimation of $\boldsymbol{\beta}$ and the corresponding error in an autoregressive framework. The least squares estimate of $\boldsymbol{\beta}$ and the associated error are defined as

$$\boldsymbol{\beta}_{\text{opt}} = \left( \boldsymbol{X}^T \boldsymbol{X} \right)^{-1} \boldsymbol{X}^T \boldsymbol{Y}, \quad E_{\text{orig}} = \| \boldsymbol{Y} - \boldsymbol{X}\boldsymbol{\beta}_{\text{opt}} \|^2. \tag{6}$$

Now, we shift to the context-aware case. Let $\boldsymbol{C}$ be an $n \times q$ matrix representing the contextual metadata corresponding to $\boldsymbol{X}$. Our key intuition is that a straightforward way to integrate this contextual information into an AR model without altering its structure is by performing an exogenous regression on the residuals. This approach preserves the original autoregressive framework, allowing the contextual information to account for the variance unexplained by the AR model. In this method, the AR model first captures the time dependencies, and the metadata refines the forecast by reducing the residual error, leading to improved accuracy. The new context-aware autoregressive model can be expressed as $\boldsymbol{Y}' = \boldsymbol{C}\boldsymbol{\gamma} + \boldsymbol{\epsilon}'$,

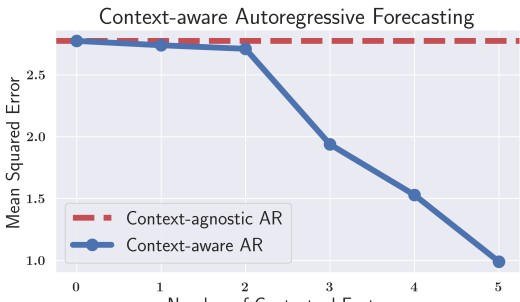

Figure 4: **Adding context improves the forecasting accuracy of an AR model**. In this experiment, we vary the number of contextual features from 0 to 5 to demonstrate how the inclusion of these features reduces the MSE for a simple Autoregressive forecaster.

where $\boldsymbol{Y}' = \boldsymbol{Y} - \boldsymbol{X}\boldsymbol{\beta}_{\text{opt}}$ represents the residuals from the AR model, $\boldsymbol{\gamma} \in \mathbb{R}^q$ is the vector of coefficients for the contextual metadata, and $\boldsymbol{\epsilon}'$ is the new error term. For this model, the least squares estimate of $\boldsymbol{\gamma}$ and the regression error is represented by

$$\boldsymbol{\gamma}_{\text{opt}} = (\boldsymbol{C}^T \boldsymbol{C})^{-1} \boldsymbol{C}^T \boldsymbol{Y}', \qquad E_{\text{new}} = \| \boldsymbol{Y}' - \boldsymbol{C}\boldsymbol{\gamma}_{\text{opt}} \|^2. \tag{7}$$

We can demonstrate that the error $E_{\text{new}}$ for the context-aware model is less than or equal to the error $E_{\text{orig}}$ of the original model. The error for this model can be expressed as

$$E_{\text{new}} = \min_{\boldsymbol{\gamma}} \| \boldsymbol{Y} - \boldsymbol{X}\boldsymbol{\beta}_{\text{opt}} - \boldsymbol{C}\boldsymbol{\gamma} \|^2. \tag{8}$$

Since $\boldsymbol{0}_q$, the zero vector in $\mathbb{R}^q$, is a feasible solution for $\boldsymbol{\gamma}$, we have

$$E_{\text{new}} = \| \boldsymbol{Y} - \boldsymbol{X}\boldsymbol{\beta}_{\text{opt}} - \boldsymbol{C}\boldsymbol{\gamma}_{\text{opt}} \|^2 \leq \| \boldsymbol{Y} - \boldsymbol{X}\boldsymbol{\beta}_{\text{opt}} \|^2 = E_{\text{orig}}. \tag{9}$$

The inequality holds because $\boldsymbol{\gamma}_{\text{opt}}$ minimizes $\| \boldsymbol{Y} - \boldsymbol{X}\boldsymbol{\beta}_{\text{opt}} - \boldsymbol{C}\boldsymbol{\gamma} \|^2$. This shows that a context-aware autoregressive forecaster is guaranteed to match or exceed the performance of its base model. Additionally, our key intuition here is that a context-aware model with zeroed coefficients for the contextual features ($\boldsymbol{\gamma} = \boldsymbol{0}_q$) will perform identically to its context-agnostic counterpart, ensuring no degradation in performance when the context is not useful.

To empirically support our theoretical justification, we conducted an experiment where samples were generated from variables dependent on their past values and underlying contextual factors. We modeled the sequences using an AR(10) process, varying the number of contextual variables $q$ from 1 to 5. Fig. 4 illustrates how incorporating contextual information improves forecasting accuracy of autoregressive model. Additional details about this experiment can be found in Appendix D.1.

Figure 5: **ContextFormer Architecture.** The architecture incorporates the multimodal contextual information in the form of metadata and temporal embeddings through a cross-attention-based method to improve the performance of an existing forecaster. During the fine-tuning phase of ContextFormer, the base model remains frozen, with only the final layer and newly added components being trained on paired contextual information.

## 5 METHODOLOGY

In this section, we propose our method, ContextFormer, to incorporate multimodal information into deep-learning forecasting models to enhance forecasting accuracy. Additionally, we propose a plug-and-play fine-tuning approach for this architecture to optimize its performance further.

### 5.1 MODEL STRUCTURE

The ContextFormer architecture, illustrated in Fig. 5, consists of a metadata embedding module, a temporal embedding module, and multiple cross-attention blocks. These components complement the base model architecture. The working of these components and the required pre-processing steps for time series forecasting are described herein:

**Base Model.** The base model for the ContextFormer can be any forecasting model capable of processing input time series and generating a hidden state representation. Neural network-based forecasters usually have an input layer, some hidden layers, and a final projection layer. The input layer processes the time series to generate embeddings, which are passed through the hidden layers. The projection layer maps the final hidden state to the dimensionality of the output $\hat{X}_{\text{future}}$.

**Metadata Embedding.** The metadata embedding block processes the paired metadata for a given time series sample $x^i \in \mathbb{R}^F$. As described in Sec. 3, the paired metadata $c^i$ comprises both categorical and continuous features denoted by $c_{\text{cat}}^i$ and $c_{\text{cont}}^i$, respectively. For ease of processing, we represent the categorical features through one-hot encoding. These categorical and continuous features are initially passed through separate dense encoders tailored to their respective types, producing individual embeddings. The resulting embeddings are then concatenated and fed into a transformer network (Vaswani et al., 2017), enabling the model to effectively capture and leverage correlations across categorical and continuous domains while respecting their distinct characteristics.

**Temporal Embedding.** Similar to the metadata embedding block, this module generates temporal embeddings from the timestamps of a given sample. Timestamps are first decomposed into components such as year, month, day, hour, and minute, depending on the dataset's granularity. These components are processed through a transformer network to extract temporal embeddings in a way similar to metadata processing. These embeddings help the model capture long-range correlations and periodic patterns within the time series.

**Cross-attention Layers.** The cross-attention layers are transformer blocks that use the hidden state representations of the historical time series, along with either the temporal or metadata embeddings, to extract relevant contextual information for forecasting.

Further details on the architectural implementation of the base models and the ContextFormer additions are provided in the Appendix D.2, with the information on design parameters, including all the time-series and metadata embedding dimensions given in Table 8 and Table 9, respectively.

## 5.2 TRAINING

The ContextFormer architecture can either be fully trained from scratch along with the base model using paired contextual metadata or can be used to fine-tune a pre-trained base model. One potential fine-tuning strategy involves a plug-and-play approach, where the context-aware model builds on a pre-trained forecaster that has already been trained on historical time-series data. In this approach, the pre-trained base model (except for the final layer) is frozen, and the ContextFormer components are added with their weights initialized to zero. The zero-initialization approach is motivated by the AR example in Sec. 4.2, where the model with zero-initialized parameters performs identically to the context-agnostic model. Subsequently, the temporal and metadata embedding modules, along with the cross-attention blocks and final projection layers, are trained for a specified number of epochs.

**Why do we opt for fine-tuning rather than training the context-aware model from scratch?**
The advantages of using the plug-and-play fine-tuning setup with a pre-trained forecaster, compared to training a context-aware model from scratch, are as follows:

1. The fine-tuned model is guaranteed to perform at least as well as the context-agnostic base model, provided the test distribution matches the training distribution. However, this guarantee does not apply to a context-aware model trained from scratch.
2. For datasets with irrelevant metadata, training a context-aware model from scratch can be unstable, potentially hindering the model from effectively learning the time series' trend and seasonality. In contrast, fine-tuning can utilize a pre-trained model, which has already captured time series dependencies, allowing it to focus solely on learning the contextual information.
3. If a time series dataset includes metadata for only some data points, training a context-aware model from scratch would either require ignoring data points without metadata or augmenting new metadata, both of which are undesirable. With our fine-tuning approach, the model can be pre-trained on the entire dataset and then fine-tuned using only the data points that have metadata.
4. The plug-and-play design of our model allows the creation of multiple context-aware models from a single forecaster. This flexibility motivates the development of dataset-agnostic universal forecasting models, which can be fine-tuned to generate dataset-specific models.

## 6 EXPERIMENTS

We have extensively evaluated the ContextFormer framework using two state-of-the-art transformer-based forecasters, PatchTST (Nie et al., 2023) and iTransformer (Liu et al., 2024), across various forecasting applications and time horizons. These evaluations showcase the impact of incorporating contextual metadata to enhance forecast accuracy. Although transformer-based forecasters were utilized as the base models in our study, the ContextFormer method is highly versatile and not limited to any particular model architecture. Its flexible design allows it to be integrated with any pre-existing forecasting model, irrespective of its internal implementation.

| MODEL | METHOD | MAE | MSE |
|---|---|---|---|
| PATCHTST | CONTEXT-AGNOSTIC | 0.749 | 1.076 |
| | CONTEXT-AWARE (OURS) | **0.702** | **0.968** |
| ITRANSFORMER | CONTEXT-AGNOSTIC | 0.764 | 1.118 |
| | CONTEXT-AWARE (OURS) | **0.704** | **0.971** |

Table 1: **Context-aware forecasters outperform Context-agnostic models on the synthetic dataset.** An average of **11.6%** improvement in MSE over baseline is observed on the ContextFormer fine-tuning of both the transformer models on the synthetic data.

**Preliminary Experiment.** Before experimenting with real-world data, we validated our architectural implementation using a synthetic dataset. In the first experiment with the ContextFormer architecture, we generated a dataset from samples of ARMA(2,2) processes with randomly chosen coefficients and added Gaussian noise to perturb the sequences (see Appendix C.1 for more details). Though the time-domain variations were minor, ARMA decomposition of the perturbed sequences revealed a significant divergence between the estimated and original generating coefficients.

The initial ARMA coefficients, treated as latent variables, represented the true underlying structure of each series but were unrecoverable from the noisy data. For our context-aware forecasting task, we utilized these latent coefficients as metadata. These coefficients were continuous and time-invariant for each series and provided critical information that could potentially enhance forecasting

| MODEL | | PATCHTST | | | ITRANSFORMER | | | BASELINES | | | |
|---|---|---|---|---|---|---|---|---|---|---|---|---|
| METHOD | | CONTEXT-AGNOSTIC | | CONTEXT-AWARE | | CONTEXT-AGNOSTIC | | CONTEXT-AWARE | | TiDE | | TIMEXER | |
| DATASET | T | MAE | MSE | MAE | MSE | MAE | MSE | MAE | MSE | MAE | MSE | MAE | MSE |
| AIR QUALITY | 48 | 0.573 | 0.770 | **0.524** | **0.674** | 0.577 | 0.771 | **0.540** | **0.696** | 0.533 | 0.682 | 0.522 | 0.659 |
| | 96 | 0.622 | 0.901 | **0.572** | **0.802** | 0.631 | 0.919 | **0.591** | **0.813** | 0.590 | 0.822 | 0.570 | 0.770 |
| ELECTRICITY | 48 | 0.038 | 0.058 | **0.029** | **0.036** | 0.042 | 0.067 | **0.028** | **0.035** | 0.030 | 0.038 | 0.029 | 0.042 |
| | 96 | 0.031 | 0.040 | **0.024** | **0.027** | 0.038 | 0.055 | **0.024** | **0.028** | 0.023 | 0.029 | 0.025 | 0.030 |
| TRAFFIC | 48 | 1.101 | 3.527 | **0.865** | **2.922** | 1.022 | 3.265 | **0.848** | **2.868** | 1.045 | 3.178 | 0.912 | 3.058 |
| | 96 | 1.084 | 3.415 | **0.845** | **2.767** | 1.025 | 3.165 | **0.830** | **2.766** | 0.967 | 2.916 | 0.901 | 2.899 |
| RETAIL | 48 | 0.123 | 0.238 | **0.115** | **0.228** | 0.129 | **0.257** | 0.124 | 0.257 | 0.124 | 0.245 | 0.123 | 0.237 |
| | 96 | 0.139 | 0.291 | **0.128** | **0.265** | 0.145 | **0.309** | 0.143 | 0.310 | 0.136 | 0.282 | 0.136 | 0.283 |
| BITCOIN | 48 | 0.854 | 1.231 | **0.821** | **1.192** | 0.832 | 1.177 | **0.810** | **1.153** | 0.790 | 1.102 | 0.964 | 1.434 |
| | 96 | 0.971 | 1.561 | **0.948** | **1.537** | 0.992 | 1.650 | **0.951** | **1.547** | 0.974 | 1.655 | 1.079 | 2.081 |
| ETT | 48 | 0.267 | 0.131 | **0.243** | **1.118** | 0.281 | 0.143 | **0.242** | **0.116** | 0.240 | 0.118 | 0.257 | 0.126 |
| | 96 | 0.299 | 0.162 | **0.281** | **0.148** | 0.313 | 0.173 | **0.280** | **0.147** | 0.283 | 0.150 | 0.286 | 0.151 |

Table 2: **ContextFormer enhances forecasting accuracy.** We compare the PatchTST and iTransformer models, with and without the ContextFormer additions for time series forecasting on the specified datasets, with a fixed lookback length of $L = 96$ and forecast horizon of $T \in \{48, 96\}$. The best results for each of the base architectures in each row are highlighted in **bold**, and the overall best results are underlined. Notably, the ContextFormer-enhanced models consistently surpass their context-agnostic counterparts across all rows and evaluation metrics. Furthermore, these models demonstrate superior performance compared to the context-aware baselines like TiDE and TimeXer in the majority of experiments.

accuracy. Since the synthetic data lacked timestamps, this experiment did not employ temporal embeddings. As shown in Table 1, the preliminary results were promising, with the inclusion of contextual information significantly improving forecasting accuracy for both architectures. Encouraged by these findings, we extended our experiments to a wide range of real-world datasets.

**Real-world Datasets.** We have validated our proposed ContextFormer approach across five forecasting tasks, each from a different domain, using a diverse group of datasets. These include the PEMS-SF (**Traffic**) and Electricity Transformer Temperature (**ETT**) dataset (specifically `ETTm2`) used in Wu et al. (2021), the ECL (**Electricity Load**) dataset from Trindade (2015), the Beijing AQ (**Air Quality**) dataset from Chen (2019), the Store Sales Competition (**Retail**) dataset from Kaggle (2022), and the Monash (**Bitcoin**) dataset from Godahewa et al. (2021) . Some of these datasets, such as Monash, ETT, and Store Sales, feature complex time-varying metadata, including online search trends and daily oil prices. On the other hand, datasets like ECL and PEMS-SF primarily contain discrete, time-invariant metadata. Additional details about the datasets and their metadata can be found in Appendix C.

**Metrics and Baselines.** The context-aware models were evaluated for forecasting on the aforementioned datasets, using a lookback length of $L = 96$ and forecasting horizons of $T = 48$ and $T = 96$. Forecast accuracy was evaluated using Mean Squared Error (MSE) and Mean Absolute Error (MAE) as performance metrics. The ContextFormer-enhanced models were benchmarked against their respective base architectures and state-of-the-art context-aware forecasters like TimeXer and TiDE. Detailed descriptions of these models and the inference procedures are provided in Appendix D. Additionally, a comparative analysis of our models with the zero-shot performance of two time-series foundational models, namely Chronos and TimesFM, is included in the Appendix E.2.

Our experiments show that incorporating contextual information into pre-trained, context-agnostic forecasters substantially improves baseline models' performance in forecasting time series data across various domains. Specifically, our experiments explore the following key questions:

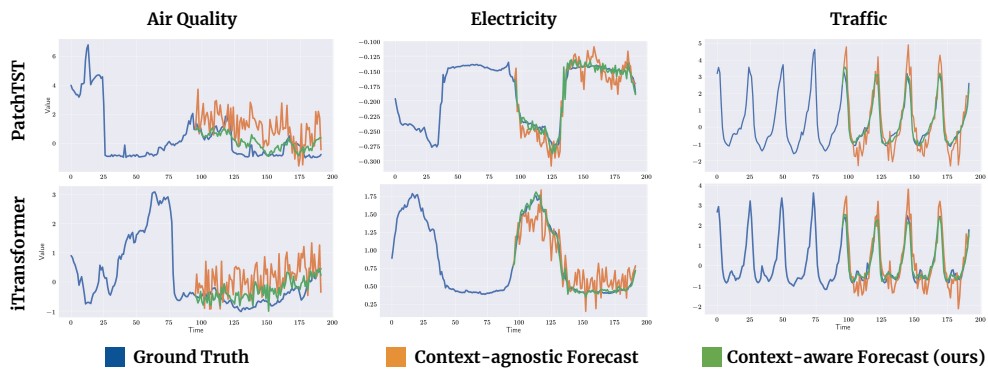

Figure 6: **Context-aware forecasts show significant qualitative improvements over context-agnostic forecasts.** The fine-tuned ContextFormer models produce forecasts that more accurately align with the ground truth distribution, offering better performance compared to the context-agnostic models.

**Does incorporating contextual metadata improve forecasting accuracy?**

The context-aware models consistently outperform their respective base architectures on both forecasting metrics across all datasets and forecasting horizons, as shown in Table 2, thereby validating our initial hypothesis. The consistent improvement across datasets highlights the ability of our approach to effectively utilize contextual information from diverse multimodal sources. The most significant gain is seen for electric load forecasting, where incorporating the metadata leads to an average improvement of **42.1%** in MSE and **28.1%** in MAE across models and forecasting horizons.

**Is this improvement in forecasting independent of the underlying base architecture?**

We used two of the most advanced transformer-based forecasting architectures for our analysis: PatchTST and iTransformer. Despite significant differences in their implementations, the addition of ContextFormer consistently improved the forecasting accuracy for both models across all experiments. This suggests that our technique has the potential to improve the performance of any transformer-based forecaster, regardless of its internal architecture. Incorporating ContextFormer modules improved MSE by an average of **13.9%** for PatchTST and **15.5%** for iTransformer.

**Is the plug-and-play fine-tuning more effective than training an entire model from scratch?**

One of our initial hypotheses was that a fine-tuned model should perform at least as well as the base model (at least on the training set), whereas no such assurance could be made for a context-aware model trained from scratch. The first part of our claim is supported by the results in Table 2, while the second part is evidenced by the values in Table 3. Here, the fully

| Training Type | Context | MAE | MSE |
|---|---|---|---|
| Base Model | Agnostic | 0.139 | 0.291 |
| Full Training | Aware | 0.154 | 0.370 |
| Fine-Tuning (Ours) | Aware | **0.128** | **0.265** |

Table 3: **Plug-and-play fine-tuning outperforms full-training.** The results for retail forecasting with $T = 96$ using the PatchTST base model show that a context-aware model trained from scratch performs worse than the context-agnostic model. In contrast, the ContextFormer model, fine-tuned in a plug-and-play manner, outperforms both.

trained context-aware model performs significantly worse than the context-agnostic model for retail. This may be caused by either unstable training or base architecture's intrinsic limitations in simultaneously learning the time series features along with the contextual correlations. In contrast, our ContextFormer mechanism is not constrained by these limitations. A more comprehensive result comparing the training methods across multiple datasets is provided in Appendix E.3.

**Can context-aware forecasting effectively capture both complex and simple metadata?**

While the air quality, store sales, and Bitcoin datasets are rich in multimodal, continuous, time-varying metadata, the traffic and electricity datasets contain only one-dimensional, discrete metadata in the form of sensor IDs and user IDs, respectively (in addition to temporal information). Our method improves performance on both datasets, demonstrating its ability to capture complex, high-dimensional metadata while leveraging temporal information and basic contextual features to boost performance. The average improvement in MSE using the complex metadata for air quality forecasting was **11.1%**, while the inclusion of temporal features and sensor IDs enhanced the average MSE for traffic forecasting by **15.2%**.

**Can the ContextFormer-enhanced models outperform SOTA context-aware forecasters ?**
Over the recent years, numerous context-aware models have been developed that incorporate co-variates as inputs while claiming superior performance compared to the transformer-based context-agnostic architectures used in our study (Wang et al., 2024b; Das et al., 2024). This claim is supported by our results in Table 2, where both TiDE and TimeXer outperform the context-agnostic models in **75%** of these experiments based on the MSE metric. On the other hand, this result gets flipped when we compare these models with the new ContextFormer-enhanced architectures; In **9** out of **12** experiments, our best-performing ContextFormer-enhanced model surpasses the performance of both the context-aware baselines, further highlighting the effectiveness of this technique in building new context-aware models from pre-trained forecasters. Additionally, note that the performance of our ContextFormer-enhanced models is limited by the constraints of base architectures; thus, with the availability of more accurate forecasters in the future, this technique could outperform any existing models that natively support covariates.

**What types of metadata modalities can be utilized by a context-aware forecaster?**
In our experiments, we incorporated diverse metadata types, including variables such as temperature, oil prices, geographic location, and web search trends. To further assess ContextFormer's ability to handle multimodal metadata, we conducted experiments forecasting Bitcoin prices using financial news articles as metadata. Since no existing dataset provided this particular combination of data, we curated a novel dataset comprising hourly Bitcoin prices from 2022 to 2024, alongside daily financial news articles scraped from the web. To incorporate the textual metadata into the forecasting model, the articles were represented as 1536-dimensional embeddings generated using the OpenAI Embeddings model (details in Appendix C). The forecasting task involved predicting Bitcoin prices one day ahead based on the previous four days of data and corresponding daily news ($L = 96, T = 24$). Results shown in Table 4 demonstrate that incorporating news articles as contextual information significantly improved forecasting metrics across both architectures, highlighting the model's effectiveness in managing complex multimodal metadata. Unlike the results for the other experiments, which are presented on a normalized scale for cross-dataset comparison, Table 4 displays the results in their original scale to emphasize the real-world economic impact of the improved forecast.

| MODEL | METHOD | MAE ($) | RMSE ($) |
|---|---|---|---|
| PATCHTST | CONTEXT-AGNOSTIC | 627.41 | 913.57 |
| | CONTEXT-AWARE (OURS) | **551.81** | **810.15** |
| ITRANSFORMER | CONTEXT-AGNOSTIC | 514.89 | 769.11 |
| | CONTEXT-AWARE (OURS) | **467.19** | **703.05** |

Table 4: **Forecasting Bitcoin prices with textual news articles.** An average of **11.5%** improvement in MAE over baseline is observed on using news articles for bitcoin price forecasting.

**Limitations.** One of the key limitations of our approach is the lack of a principled method to find which metadata or contextual feature is important for forecasting. Identifying the key metadata features in advance can help limit the diversity of metadata provided as input to the model, thereby simplifying the learning process during the fine-tuning stage.

## 7 CONCLUSION

In this paper, we introduced ContextFormer, a novel technique for integrating contextual information into existing forecasting models. Through comprehensive evaluations on diverse real-world datasets, we demonstrated ContextFormer's ability to effectively handle complex, multimodal metadata while consistently outperforming baseline models and even forecasting foundation models. In addition, we proposed a resource-efficient plug-and-play fine-tuning framework that offers significant improvements to forecasting accuracy over training context-aware models from scratch.

In future work, we aim to test our approach on other contextual modalities such as images, videos, etc. An interesting future work is to analyze the effects of forecasting metadata first; thereby, we propose a two-step forecasting pipeline where we first forecast metadata. We hypothesize that the forecasted metadata can be used to obtain better forecasts. Our key intuition is that metadata is more human-interpretable, and therefore forecasting metadata could be an easier task to solve.

**Reproducibility.** The implementation and hyperparameter details have been provided to help reproduce the results reported in the paper. The source code will be released post publication.

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

# Appendix

## A  MSE Loss and Mutual Information

Following the approach in Jing et al. (2022), we can demonstrate that minimizing the MSE loss between $\boldsymbol{X}_{\text{future}}$ and $\hat{\boldsymbol{X}}_{\text{future}}$ is equivalent to maximizing the mutual information $\mathcal{I}\left(\boldsymbol{X}_{\text{future}}; \hat{\boldsymbol{X}}_{\text{future}}\right)$, assuming a Gaussian noise model between the two variables. Specifically, minimizing the MSE is shown to be equivalent to maximizing the log-likelihood, which, in turn, maximizes the mutual information.

Suppose that the relationship between $\boldsymbol{X}_{\text{future}}$ and $\hat{\boldsymbol{X}}_{\text{future}}$ is modeled as

$$\hat{\boldsymbol{X}}_{\text{future}} = \boldsymbol{X}_{\text{future}} + Z,$$

where $Z$ is Gaussian noise with zero mean and variance $\sigma^2$, i.e., $Z \sim \mathcal{N}(0, \sigma^2)$. The log-likelihood of obtaining $\boldsymbol{X}_{\text{future}}$ given $\hat{\boldsymbol{X}}_{\text{future}}$ can be derived from the probability density function of the Gaussian distribution. The log-likelihood function is given by

$$\log p(\boldsymbol{X}_{\text{future}} \mid \hat{\boldsymbol{X}}_{\text{future}}) = -\frac{1}{2} \log(2\pi\sigma^2) - \frac{\|\boldsymbol{X}_{\text{future}} - \hat{\boldsymbol{X}}_{\text{future}}\|_2^2}{2\sigma^2}.$$

Since $\frac{1}{2}\log(2\pi\sigma^2)$ is a constant with respect to $\boldsymbol{X}_{\text{future}}$, the log-likelihood is maximized when the term $\|\boldsymbol{X}_{\text{future}} - \hat{\boldsymbol{X}}_{\text{future}}\|_2^2$ is minimized, which is the same as minimizing the MSE. Therefore, *minimizing the MSE is equivalent to maximizing the log-likelihood.*

Having shown the equivalence of MSE and the log-likelihood, now the mutual information $\mathcal{I}(\boldsymbol{X}_{\text{future}}; \hat{\boldsymbol{X}}_{\text{future}})$ between $\boldsymbol{X}_{\text{future}}$ and $\hat{\boldsymbol{X}}_{\text{future}}$ can be expressed as

$$\mathcal{I}(\boldsymbol{X}_{\text{future}}; \hat{\boldsymbol{X}}_{\text{future}}) = \mathcal{H}(\boldsymbol{X}_{\text{future}}) - \mathcal{H}(\boldsymbol{X}_{\text{future}} \mid \hat{\boldsymbol{X}}_{\text{future}})$$

where $\mathcal{H}(\boldsymbol{X}_{\text{future}})$ is the entropy of $\boldsymbol{X}_{\text{future}}$ and $\mathcal{H}(\boldsymbol{X}_{\text{future}} \mid \hat{\boldsymbol{X}}_{\text{future}})$ is the conditional entropy of $\boldsymbol{X}_{\text{future}}$ given $\hat{\boldsymbol{X}}_{\text{future}}$. For Gaussian noise, $\mathcal{H}(\boldsymbol{X}_{\text{future}} \mid \hat{\boldsymbol{X}}_{\text{future}})$ is related to the conditional variance of $\boldsymbol{X}_{\text{future}}$ given $\hat{\boldsymbol{X}}_{\text{future}}$,

$$\mathcal{H}(\boldsymbol{X}_{\text{future}} \mid \hat{\boldsymbol{X}}_{\text{future}}) = -\mathbb{E}\left[\log p(\boldsymbol{X}_{\text{future}} \mid \hat{\boldsymbol{X}}_{\text{future}})\right].$$

Maximizing the likelihood of the observed data $\hat{\boldsymbol{X}}_{\text{future}}$ given the model (in this case, $\boldsymbol{X}_{\text{future}}$) reduces the uncertainty $\mathcal{H}(\boldsymbol{X}_{\text{future}} \mid \hat{\boldsymbol{X}}_{\text{future}})$, effectively increasing the mutual information. Now, we know that minimizing the MSE maximizes the log-likelihood. This corresponds to making the estimate $\hat{\boldsymbol{X}}_{\text{future}}$ as close as possible to the true value $\boldsymbol{X}_{\text{future}}$, which reduces the variance of the noise $Z$ (or the uncertainty in $\hat{\boldsymbol{X}}_{\text{future}}$).

Since mutual information $\mathcal{I}(\boldsymbol{X}_{\text{future}}; \hat{\boldsymbol{X}}_{\text{future}})$ is a measure of the reduction in uncertainty about $\boldsymbol{X}_{\text{future}}$ given $\hat{\boldsymbol{X}}_{\text{future}}$, minimizing the conditional variance (or equivalently, maximizing the log-likelihood) increases $\mathcal{I}(\boldsymbol{X}_{\text{future}}; \hat{\boldsymbol{X}}_{\text{future}})$. Thus, *minimizing MSE maximizes the log-likelihood, which in turn maximizes the mutual information $\mathcal{I}(\boldsymbol{X}_{\text{future}}; \hat{\boldsymbol{X}}_{\text{future}})$,*

$$\min \mathbb{E}_{p_{\text{data}}} \|\boldsymbol{X}_{\text{future}} - \hat{\boldsymbol{X}}_{\text{future}}\|_2 \iff \max \mathcal{I}\left(\boldsymbol{X}_{\text{future}}; \hat{\boldsymbol{X}}_{\text{future}}\right).$$

## B  TimesFM with Covariates

**TimesFM** (Das et al., 2024), developed by Google Research, is one of the latest foundational models for time series forecasting. It boasts superior zero-shot performance compared to other foundational models across most of the commonly used benchmark datasets. A key feature of TimesFM is its ability to incorporate static and dynamic covariates during inference, making it a context-aware forecaster by our definition. As a multipurpose model, it is not trained on any dataset-specific covariates but at the time of dataset-specific inference, it treats them as exogenous regression variables and fits linear models onto them.

One limitation of such a simplistic batched in-context regression model is that it may be incapable of extracting complex correlations among the covariates and the time series. Moreover, the TimesFM implementation requires the presence of future values of dynamic covariates through the forecasting horizon; this kind of information is often unavailable in real-world scenarios. To address this, the TimesFM developers have proposed two stop-gap solutions: either shifting and repeating past dynamic covariates as delayed proxies for the future or "bootstrapping", where TimesFM is first used to forecast these past covariates into the future and then called again using these forecasts as future covariates. We employ the earlier method to evaluate the model's ability to perform using only historical covariates (or contextual information, in our terms). The results, given in table 5, highlight the model's inability to improve its performance via its current context-aware implementation in the absence of future covariates, as for all of our datasets (except one), the context-agnostic TimesFM outperforms the context-aware one on both the metrics.

| DATASET | T | CONTEXT-AGNOSTIC | | CONTEXT-AWARE | |
|---|---|---|---|---|---|
| | | MAE | MSE | MAE | MSE |
| AIR QUALITY | 48 | **0.571** | **0.807** | 0.611 | 0.847 |
| | 96 | **0.638** | **0.986** | 0.659 | 1.01 |
| ELECTRICITY | 48 | **0.073** | **0.156** | 0.084 | 0.249 |
| | 96 | **0.057** | **0.129** | 0.065 | 0.188 |
| TRAFFIC | 48 | **0.702** | **2.172** | 0.795 | 2.516 |
| | 96 | **0.679** | **2.143** | 0.758 | 2.428 |
| RETAIL | 48 | **0.108** | **0.185** | 0.114 | 0.215 |
| | 96 | 0.133 | **0.236** | **0.131** | 0.274 |
| BITCOIN | 48 | **0.667** | **0.822** | 0.724 | 0.921 |
| | 96 | **0.952** | **1.509** | 0.973 | 1.572 |

Table 5: **TimesFM's inability to leverage historical metadata through in-context regression.** Without access to future covariate values, the context-aware TimesFM model does not demonstrate any performance improvement across the datasets used in our main experiment using its simplistic linear regression approach.

## C DATASET DESCRIPTION

We will now describe the datasets used for training, validation and evaluation of our proposed model.

### C.1 SYNTHETIC DATA

We generated a dataset comprising 10000 time-series sequences, each containing 192 samples, based on ARMA(2,2) processes with randomly initialized coefficients, sampled from a uniform distribution, for the first preliminary experiment. The stability of each ARMA process was ensured by verifying that all the roots of the corresponding characteristic equations were within the unit circle. To cause some perturbation, Gaussian noise with 0.1 variance was added to all the sequences. The metadata for the dataset, consisting of the four ARMA coefficients, was continuous and time-invariant. These coefficients differed significantly from the ARMA coefficients obtained from the noisy data for most of the sequences. The dataset was split in a 7:1:2 ratio among the train, validation, and test splits.

### C.2 BEIJING AQ (AIR QUALITY)

The dataset obtained from Chen (2019) contains hourly air pollutant concentration data and corresponding meteorological data from 12 locations in Beijing. The task is to forecast a 6-channel multivariate time series using historical data and weather forecast metadata. Missing values in the dataset are handled by imputing continuous metadata and time series values with their mean, while missing categorical metadata (such as wind direction) are assigned an "unknown" label. The data, spanning from 2013 to 2017, is split into training, validation, and test sets in a 7:1:2 ratio. For each set, we first apply a sliding window of length 144 with a stride of 24, resulting in 9828 training, 1332 validation, and 2796 test time series samples. Then, we use a sliding window of length 192 with a stride of 24, yielding 9012 training, 1188 validation, and 2556 test time series.

### C.3 STORE SALES (RETAIL)

The dataset, sourced from a popular Kaggle competition (Kaggle, 2022), contains daily sales data from 2013 to 2017 for 34 product families sold across 55 Favorita stores in Ecuador. The dataset includes features such as store number, product family, promotional status, and sales figures. Additionally, supplementary information like store metadata and oil prices is provided, offering time-

varying metadata that can be leveraged for forecasting. For the forecasting tasks, we consider the complete time series for each product in each store, which is univariate time series. Initially, we apply a sliding window of length 144 with a stride of 24, resulting in 53460 training, 17820 validation, and 24948 test time series samples. Next, we use a sliding window of length 192 with a stride of 24, yielding 49896 training, 14256 validation, and 21384 test time series samples.

## C.4 PEMS-SF (TRAFFIC)

The dataset, sourced from Wu et al. (2021), contains 15 months of daily data from the California Department of Transportation PEMS website. It captures the occupancy rate (ranging from 0 to 1) of various freeway car lanes in the San Francisco Bay area. The data spans from 2008 to 2009, with measurements sampled every hour . The forecasting task involves predicting the electricity demand pattern for a single sensor (selected from a total of 861 sensors), framed as a univariate time series forecasting problem. The data is split temporally into training, validation, and test sets in a 7:1:2 ratio. We then apply sliding windows of lengths 144 and 192, each with a stride of 24. The number of samples for all the sets is given in Table 7.

| DATASET | TIME-SERIES DATA | CONTINUOUS METADATA | CATEGORICAL METADATA |
|---|---|---|---|
| AIR QUALITY | CO, NO$_2$, SO$_2$, O$_3$, PM2.5, AND PM10 CONCENTRATION | TEMPERATURE, HUMIDITY, WIND SPEED, PRESSURE, DEW POINT | LOCATION, WIND DIRECTION |
| RETAIL | PRODUCT SALES VOLUME | PROMOTIONAL OFFERS, OIL PRICES | STORE ID, LOCATION, ITEM CATEGORY |
| TRAFFIC | TRAFFIC VOLUME | NONE | SENSOR ID |
| ELECTRICITY LOAD | ELECTRICITY CONSUMPTION | NONE | USER ID |
| ETT | OIL TEMPERATURE | 6 POWER LOAD FEATURES | NONE |
| BITCOIN | BITCOIN PRICES | 17 FACTORS | NONE |

Table 6: **Dataset Summary.** For our main experiments, we selected real-world datasets that encompass prevalent forecasting tasks across diverse domains, including environmental science, energy, finance, retail, and transportation.

## C.5 ECL (ELECTRICITY )

The dataset taken from Trindade (2015) consists of power consumption data for 370 users in Portugal over a period of 4 years from 2011 to 2015. It is a commonly used dataset for time series forecasting (Wu et al., 2021; Liu et al., 2024; Ansari et al., 2024). The forecasting task with respect to this dataset is to predict the electricity demand pattern for a single user, which is framed as a univariate time series forecasting problem. In the absence of any innate metadata features, we consider the 370 user IDs to be the only metadata. The data is sampled every 15 minutes, resulting in a time series with 96 daily timesteps. We process the data to remove days with significant 0 values. The data has been split, and sliding windows with stride 24 are used to get approximately a 7:1:2 ratio for the number of samples in the training, validation, and test sets, with the exact numbers given in table 7.

## C.6 ELECTRICITY TRANSFORMER TEMPERATURE (ETT)

The dataset originally introduced by Zhou et al. (2021) comprises oil temperature and power load factors for electricity transformers from two distinct counties in China, spanning two years. Like the ECL dataset, it is widely utilized for time series forecasting tasks (Wu et al., 2021; Liu et al., 2024). Among the various variants of the ETT dataset, we selected `ETTm2` for our experiments. The forecasting task for this dataset involves predicting the oil temperature of an electricity transformer, framed as a univariate time series forecasting problem. Six power load factors serve as covariates. The data is sampled at 15-minute intervals, resulting in a time series with 96 daily timesteps. Using a sliding window approach with a stride of 24, the dataset is split into training, validation, and test sets in approximately a 7:1:2 ratio.

### C.7 Monash (Bitcoin)

The dataset, sourced from the Monash Time Series Forecasting Repository (Godahewa et al., 2021), contains daily Bitcoin closing prices from 2010 to 2021, along with 18 potential influencing factors. These include metrics like hash rate, block size, mining difficulty, and social indicators such as the number of tweets and Google searches related to the keyword "Bitcoin." During preprocessing, we excluded the number of tweets due to its limited availability, leaving us with 17-dimensional continuous time-varying metadata for the univariate forecasting task. The dataset is divided into training, validation, and test sets in a 7:1:2 ratio. We apply sliding windows of lengths 144 and 192 with a stride of 24 to the data.

### C.8 Additional Experiment (Bitcoin-News)

In the absence of a dataset containing both Bitcoin prices and corresponding news articles, we constructed a new dataset comprising hourly Bitcoin closing prices and daily financial news articles. The articles from January 1st, 2022, to February 17th, 2024, were sourced using the Alpaca Historical News API (Alpaca, 2024), with each metadata instance consisting of all news articles and headlines tagged with `BTCUSD` for a given day. These textual instances were directly processed using the OpenAI Embeddings model 'text-embedding-1-small' (OpenAI, 2022), producing 1536-dimensional embeddings for each day's news. To ensure causality, the hourly Bitcoin closing prices for a given day were aligned with the previous day's news embeddings. These embeddings serve as time-varying metadata, remaining constant within a day but varying daily. This univariate forecasting experiment was conducted with a fixed lookback length of $L = 96$ and a horizon of $T = 24$, allowing us to forecast daily Bitcoin trends based on the previous four days' history and news articles. As before, the data was split temporally into training, validation, and test sets in a 7:1:2 ratio, with a sliding window of length 120 and a stride of 24 applied to create the datasets.

| Dataset | Channels | Size ($T=48$) | Size ($T=96$) | Frequency | Batch Size |
|---|---|---|---|---|---|
| Synthetic | 1 | - | (7000,1000,2000) | N.A. | 128 |
| Air Quality | 6 | (9828,1332,2796) | (9012,1188,2556) | Hourly | 32 |
| Retail | 1 | (53460,17820,24948) | (49896,14256,21384) | Daily | 128 |
| Electricity | 1 | (50265,9102,13627) | (50262,8750,13626) | 15 Minute | 128 |
| Traffic | 1 | (435666, 58548, 121401) | (433944, 56826, 119679) | Hourly | 384 |
| ETT | 1 | (2025,283,573) | (2027,285,575) | 15 Minute | 64 |
| Bitcoin | 1 | (104,10,26) | (102,8,24) | Daily | 1 |
| Bitcoin-News | 1 | - | (524,71,148) | Hourly | 4 |

Table 7: **Dataset Description.** This table summarizes the main features of the datasets used in our experiments, such as dimensionality, sample size, frequency, and training batch size. The dataset sample size are given in the format of (training size, validation size, test size).

## D Implementation Details

### D.1 Context-aware Autoregression

To provide theoretical justification for the ideas in Subsection 4.2, we generated a dataset consisting of 500-length sequences formed by linear combinations of an autoregressive process with five latent variables, followed by added perturbations. For this scenario, we choose the latent variables as the contextual information. The resulting dataset, denoted as $\mathcal{D}$, is structured as $\{(\boldsymbol{X}^n, \boldsymbol{C}^n, \boldsymbol{Y}^n)\}_{n=1}^{N}$, For our experiments, we set $N = 1000$.

We initially start by modeling these sequences through vanilla AR(10) models of the form $\boldsymbol{Y} = \boldsymbol{X}\boldsymbol{\beta} + \boldsymbol{\epsilon}$, where $\boldsymbol{X}$ is a $490 \times 10$ matrix and $\boldsymbol{Y}$ is a $490$-length vector. The corresponding paired metadata for such a sequence can be represented as $\boldsymbol{C}$, which is a $490 \times 5$ matrix.

Next, we incorporate contextual metadata into the existing AR models by fitting them as exogenous regressors for the residuals. To demonstrate the impact of increasing context on forecasting accuracy, we gradually increase the dimensionality of the regression model from $q = 1$ to 5. The results for this context-aware autoregressive forecaster are visualized in Fig. 4.

## D.2 Context-aware Transformers

### D.2.1 Base Architectures

The core architecture for both the ContextFormer-enhanced forecasters contains an input history embedding layer, six hidden layer blocks, and a final projection layer. Each of the hidden layer blocks consists of two parallel cross-attention layers and a self-attention layer. Each attention layer operates in a 256-dimensional representational space while employing an 8-head attention mechanism.

**PatchTST** (Nie et al., 2023) is a popular transformer-based architecture for time series forecasting. In contrast to the traditional models that treat each time step as an individual token, PatchTST divides the data into patches, similar to what Vision Transformers (Dosovitskiy et al., 2021) do for images. Each patch in this setup represents a sequence of time steps, which enables the model to focus on long-term temporal patterns. By applying self-attention to these patches, PatchTST captures long-range dependencies more efficiently, reducing the computational cost associated with traditional transformers. This patch-based approach enables the model to handle longer sequences and large-scale forecasting tasks more effectively. PatchTST outperforms standard transformers on various benchmarks and can handle both univariate and multivariate time series data, making it a versatile choice for various forecasting tasks.

**iTransformer** (Liu et al., 2024) extends the patching concept introduced in PatchTST by applying the model to inverted dimensions of the time series. Rather than embedding time steps, iTransformer treats each variable or feature of the time series as separate tokens. This shifts the focus from temporal dependencies to relationships between features across time. Despite this inversion, iTransformer retains core Transformer components, including multi-head attention and positional feed-forward networks, but applies them in a way that fundamentally alters how dependencies are modeled. Although the original iTransformer architecture can handle timestamps, we intentionally excluded them for the context-agnostic variant of the model. The rest of the implementation for both the base models is the same as what is given in the official TimesNet (Wu et al., 2022) Github repository.

| Design Parameter | Value |
|---|---|
| Embedding Dimension | 256 |
| Self-attention Layers | 6 |
| Attention Heads | 8 |
| Activation | GeLU |
| Patch Length | 16 |
| Stride | 8 |
| Learning Rate | $3 \times 10^{-5}$ |
| Dropout | 0.1 |

Table 8: **Design parameters for our experiments.** The hyperparameters for embedding dimensions, attention heads, and activation functions are consistent across all the transformers in the base model and ContextFormer additions. The number of self-attention layers is the same for both the base models. The learning rate and dropout remain fixed throughout all the forecasters across the experiments. The parameters for patch length and stride are specific to the PatchTST input layer.

### D.2.2 ContextFormer Additions

**Metadata Embedding.** The metadata embedding module consists of two fully connected (FC) networks followed by a transformer encoder. Discrete metadata is one-hot encoded and processed through one FC network, while continuous metadata is passed through a separate FC network. The input size of each network corresponds to the number of discrete or continuous features in the dataset, with the output dimensions consistently set to 256 for all experiments. If both metadata types are present, then their outputs are concatenated and processed through a linear layer to reduce the dimensionality from 512 to 256; otherwise, the single output is used. The resulting output is then added to the positional encodings and passed through a transformer encoder, generating the metadata embedding. This implementation of metadata embedding follows the approach described in Narasimhan et al. (2024) for constructing metadata encoders in conditional time series generation.

| Design Parameter | Value |
|---|---|
| Feed-forward Dimension | 256 |
| Embedding Dimension | 256 |
| Self-attention Layers | 2 |
| Attention Heads | 8 |
| Activation | GeLU |

Table 9: **Hyperparameters for the embedding modules.** Both metadata and temporal embedding blocks share the same design parameters.

**Temporal Embedding.** The temporal embedding block shares a similar architecture with the metadata embedding but is specifically designed for continuous temporal data. Timestamp information, such as the year, month, day, and hour, is decomposed based on the dataset's granularity and treated as continuous contextual features. In essence, the temporal embedding functions like a metadata embedding module, where the temporal data is embedded as continuous metadata. Unlike the metadata encoder, the temporal encoder exclusively handles continuous features and focuses entirely on capturing the temporal characteristics of the input.

### D.2.3 OTHER DETAILS

**Training Parameters.** The learning rate was set to $3 \times 10^{-5}$ for all experiments, and a dropout rate of 0.1 was applied throughout the training process. Each of the ContextFormer models was trained for 100 epochs: the first 50 epochs focused on training the base models using historical data, while the remaining 50 epochs were dedicated to the ContextFormer fine-tuning. In the experiments described in Appendix E.3, the fully-trained context-aware models were trained on the time series data and paired metadata until convergence. The model with the lowest validation loss was saved and used for inference. All the experiments, including the benchmarks, were run on three random seeds to ensure the robust results, the Table 2 reports average values of the evaluation metrics over the random seeds.

**Model Size.** The context-aware models comprised an average of approximately 13 million parameters. Of these, an average of 3.5 million parameters were associated with the original model, while the remaining approximately 9.5 million parameters were introduced by the ContextFormer additions. The total number of parameters varied slightly depending on the specific base architecture, the dimensions of the time series, and the corresponding metadata.

**Software and Hardware.** All experiments were conducted using Python 3.9.12 and PyTorch 2.0.0 (Paszke et al., 2019), running on Nvidia RTX A5000 GPUs.

### D.3 BENCHMARK FORECASTERS

**TiDE** (Das et al., 2023), introduced by Google Research in 2023, is a straightforward MLP-based encoder-decoder architecture designed for long-term time series forecasting. It effectively handles non-linear dependencies and dynamic covariates, presenting itself as a parameter-efficient model. Unlike transformer-based solutions, TiDE avoids self-attention, recurrent, or convolutional mechanisms, achieving linear computational scaling with respect to context and horizon lengths. The model encodes the historical time-series data along with covariates using dense MLPs and decodes the series alongside future covariates, also leveraging dense MLPs. To align with our problem formulation outlined in Section 3, we adapted TiDE by masking future covariates during both training and inference. The embedding dimension was set to 16, as given in the TimeXer GitHub repository.

**TimeXer** (Wang et al., 2024b), introduced in 2024, is an innovative transformer-based architecture designed for time series forecasting that incorporates exogenous variables through a cross-attention mechanism. It utilizes patch-level representations for endogenous variables and variate-level representations for exogenous variables, linked via an endogenous global token. This architecture aims to jointly capture intra-endogenous temporal dependencies and exogenous-to-endogenous correlations. To ensure a fair comparison with our models, we maintained the design parameters as specified in Table 8. Since neither of the benchmark forecasters includes a separate processing pipeline for discrete and continuous metadata, we concatenated these metadata types and passed them jointly through the models for all benchmarking experiments. Training parameters were kept consistent with those outlined in Appendix D.2.3, while the remaining parameters were adopted directly from the official TimeXer GitHub repository.

## E ABLATION STUDY

### E.1 MAIN RESULTS WITH STANDARD DEVIATION

For robust experimental results, each experiment is repeated three times with different random seeds. Due to space constraints, the main text presents the results without standard deviations. The complete results, including standard deviations, are provided in Table 10.

| MODEL | | PATCHTST | | | | ITRANSFORMER | | | |
|---|---|---|---|---|---|---|---|---|---|
| METHOD | | CONTEXT-AGNOSTIC | | CONTEXT-AWARE | | CONTEXT-AGNOSTIC | | CONTEXT-AWARE | |
| DATASET | T | MAE | MSE | MAE | MSE | MAE | MSE | MAE | MSE |
| AIR QUALITY | 48 | $0.573 \pm 0.005$ | $0.770 \pm 0.013$ | $0.524 \pm 0.002$ | $0.674 \pm 0.011$ | $0.577 \pm 0.005$ | $0.771 \pm 0.006$ | $0.540 \pm 0.002$ | $0.696 \pm 0.002$ |
| | 96 | $0.622 \pm 0.006$ | $0.901 \pm 0.014$ | $0.572 \pm 0.004$ | $0.802 \pm 0.009$ | $0.631 \pm 0.004$ | $0.919 \pm 0.007$ | $0.591 \pm 0.006$ | $0.813 \pm 0.013$ |
| ELECTRICITY | 48 | $0.038 \pm 0.001$ | $0.058 \pm 0.003$ | $0.024 \pm 0.001$ | $0.029 \pm 0.001$ | $0.042 \pm 0.002$ | $0.067 \pm 0.008$ | $0.028 \pm 0.001$ | $0.035 \pm 0.001$ |
| | 96 | $0.031 \pm 0.005$ | $0.040 \pm 0.009$ | $0.024 \pm 0.001$ | $0.027 \pm 0.002$ | $0.038 \pm 0.001$ | $0.055 \pm 0.002$ | $0.024 \pm 0.001$ | $0.028 \pm 0.001$ |
| TRAFFIC | 48 | $1.101 \pm 0.066$ | $3.527 \pm 0.179$ | $0.865 \pm 0.009$ | $2.922 \pm 0.047$ | $1.022 \pm 0.019$ | $3.265 \pm 0.096$ | $0.848 \pm 0.016$ | $2.868 \pm 0.024$ |
| | 96 | $1.084 \pm 0.042$ | $3.415 \pm 0.173$ | $0.845 \pm 0.014$ | $2.767 \pm 0.049$ | $1.025 \pm 0.031$ | $3.165 \pm 0.034$ | $0.830 \pm 0.002$ | $2.766 \pm 0.020$ |
| RETAIL | 48 | $0.123 \pm 0.002$ | $0.238 \pm 0.005$ | $0.115 \pm 0.001$ | $0.228 \pm 0.001$ | $0.129 \pm 0.002$ | $0.257 \pm 0.004$ | $0.124 \pm 0.001$ | $0.257 \pm 0.005$ |
| | 96 | $0.139 \pm 0.002$ | $0.291 \pm 0.010$ | $0.128 \pm 0.001$ | $0.265 \pm 0.004$ | $0.145 \pm 0.002$ | $0.309 \pm 0.006$ | $0.143 \pm 0.001$ | $0.310 \pm 0.012$ |
| BITCOIN | 48 | $0.854 \pm 0.045$ | $1.231 \pm 0.120$ | $0.821 \pm 0.038$ | $1.192 \pm 0.101$ | $0.832 \pm 0.020$ | $1.177 \pm 0.053$ | $0.810 \pm 0.021$ | $1.153 \pm 0.078$ |
| | 96 | $0.971 \pm 0.004$ | $1.561 \pm 0.029$ | $0.948 \pm 0.003$ | $1.537 \pm 0.009$ | $0.992 \pm 0.010$ | $1.650 \pm 0.029$ | $0.951 \pm 0.005$ | $1.547 \pm 0.022$ |

Table 10: **Results with Standard Deviation.** We compare the PatchTST and iTransformer with and without the ContextFormer additions on forecasting time series with a fixed lookback length $L = 96$ and forecast horizon $T \in \{48, 96\}$. The mean and standard deviation are calculated for each experiment with three different seeds.

## E.2 CONTEXTFORMER VS. FOUNDATIONAL MODELS

To compare the performance of our methods with massive foundational models, we have marked them against Chronos (Ansari et al., 2024) and TimesFM Das et al. (2024). Chronos, introduced by Amazon Sciences in 2024, is a collection of pre-trained foundational time series models that leverage LLM architectures. For our experiments, we specifically use the zero-shot forecasting results from the 'chronos-t5-base' variant with 200 million parameters. Note that Chronos can handle only single-channel data; thus, for the evaluation of the multivariate datasets, all the channels were processed separately. The details for TimesFM are given in Appendix B.

| TYPE | | CONTEXTFORMER | | | | FOUNDATIONAL MODELS | | | |
|---|---|---|---|---|---|---|---|---|---|
| MODEL | | PATCHTST | | ITRANSFORMER | | CHRONOS | | TIMESFM | |
| DATASET | T | MAE | MSE | MAE | MSE | MAE | MSE | MAE | MSE |
| AIR QUALITY | 48 | **0.524** | **0.674** | 0.540 | 0.696 | 0.600 | 0.873 | 0.571 | 0.807 |
| | 96 | **0.572** | **0.802** | 0.591 | 0.813 | 0.664 | 0.989 | 0.638 | 0.986 |
| ELECTRICITY | 48 | 0.029 | 0.036 | **0.028** | **0.035** | 0.069 | 0.104 | 0.073 | 0.156 |
| | 96 | 0.024 | 0.027 | **0.024** | **0.028** | 0.058 | 0.078 | 0.057 | 0.129 |
| TRAFFIC | 48 | 0.865 | 2.922 | 0.848 | 2.868 | 0.838 | 4.845 | **0.702** | **2.172** |
| | 96 | 0.845 | 2.767 | 0.830 | 2.766 | 0.842 | 5.142 | **0.679** | **2.143** |
| RETAIL | 48 | 0.115 | 0.228 | 0.124 | 0.257 | **0.106** | 0.198 | 0.108 | **0.185** |
| | 96 | 0.128 | 0.265 | 0.143 | 0.310 | **0.122** | 0.245 | 0.133 | **0.236** |
| BITCOIN | 48 | 0.821 | 1.192 | 0.810 | 1.153 | 0.677 | 0.885 | **0.667** | **0.822** |
| | 96 | 0.948 | 1.537 | 0.951 | 1.547 | **0.898** | **1.501** | 0.952 | 1.509 |

Table 11: **ContextFormer Vs. Foundational Models.** We compare our ContextFormer-enhanced models with the zero-shot performance of Chronos and TimesFM for time series forecasting with a fixed lookback length $L = 96$ and forecast horizon $T \in \{48, 96\}$. For all the datasets, our models offer comparable or better performance with respect to the massive foundational models.

## E.3 FULL-TRAINING VS. FINE-TUNING OF CONTEXT-AWARE MODELS

To empirically support the use of fine-tuning over training from scratch, we present the results for numerous datasets using both approaches. These results in Table 12 show that even when training the context-aware architecture from scratch, its performance often falls short of our fine-tuned models for most of the datasets. This discrepancy could be attributed to the factors discussed in Sec. 5.2 or other potential causes.

| MODEL | | PATCHTST | | | | ITRANSFORMER | | | |
|---|---|---|---|---|---|---|---|---|---|
| TRAINING | | FULL | | FINE-TUNE | | FULL | | FINE-TUNE | |
| DATASET | T | MAE | MSE | MAE | MSE | MAE | MSE | MAE | MSE |
| AIR QUALITY | 48 | **0.507** | **0.611** | 0.524 | 0.674 | **0.530** | **0.646** | 0.540 | 0.696 |
| | 96 | **0.550** | **0.717** | 0.572 | 0.802 | **0.562** | **0.734** | 0.591 | 0.813 |
| ELECTRICITY | 48 | 0.030 | 0.038 | **0.029** | **0.036** | 0.029 | 0.035 | **0.028** | **0.035** |
| | 96 | **0.022** | **0.025** | 0.024 | 0.027 | 0.025 | 0.029 | **0.024** | **0.028** |
| TRAFFIC | 48 | 0.994 | 3.128 | **0.865** | **2.922** | 0.917 | 2.903 | **0.848** | **2.868** |
| | 96 | 0.955 | 3.006 | **0.845** | **2.767** | 0.896 | 2.859 | **0.830** | **2.766** |
| RETAIL | 48 | 0.116 | 0.230 | **0.115** | **0.228** | **0.121** | **0.252** | 0.124 | 0.257 |
| | 96 | 0.154 | 0.370 | **0.128** | **0.265** | 0.151 | 0.326 | **0.143** | **0.310** |
| BITCOIN | 48 | 0.849 | 1.226 | **0.821** | **1.192** | 1.038 | 1.796 | **0.810** | **1.153** |
| | 96 | 0.983 | 1.656 | **0.948** | **1.537** | 0.962 | 1.604 | **0.951** | **1.547** |

Table 12: **Full Training Vs. Fine-tuning.** We compare the two training methods for the context-aware PatchTST and iTransformer for time series forecasting with a fixed lookback length $L = 96$ and forecast horizon $T \in \{48, 96\}$. For most of the datasets, our plug-and-play fine-tuning method outperforms its fully-trained counterpart.

We further validated our choice of fine-tuning the ContextFormer-enhanced models instead of training them from scratch through the training curves. Figure 7 illustrates the training and validation losses for both full training and fine-tuning approaches in the Bitcoin Dataset. For the ContextFormer fine-tuning, the base architecture is pre-trained for 50 epochs, after which it is frozen, and the ContextFormer additions are trained for the next 50 epochs. The full training occurs for 100 epochs. From Figure 7, it is evident that the validation loss for full training begins to diverge significantly earlier than for fine-tuning. This divergence occurs well before the 50th epoch. Furthermore, the ContextFormer-enhanced PatchTST model achieves a lower best validation loss through the proposed fine-tuning strategy as compared to full training from scratch.

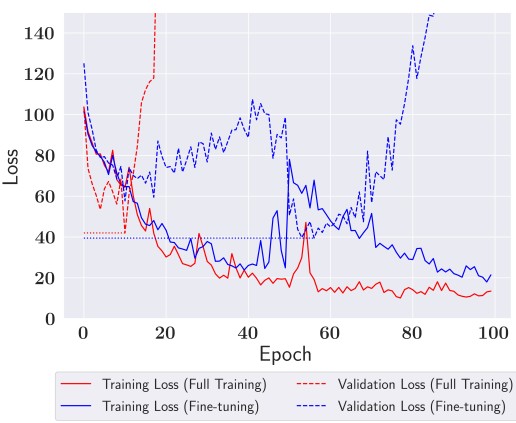

Figure 7: **Fine-tuning achieves better validation loss on the Bitcoin dataset.** Training and validation loss curves for full training and fine-tuning on the Bitcoin dataset with a forecast horizon $T = 96$, for ContextFormer-enhanced PatchTST.

### E.4 EFFECT OF TEMPORAL INFORMATION VS FULL CONTEXT

Throughout this text, we have used the term *contextual information* to encompass all static or time-varying, continuous, or discrete data associated with the time series in question. This includes easily accessible temporal information such as timestamps (e.g., month or day), which can significantly impact forecasting accuracy, especially for datasets with long-range periodicity. While frameworks like GLAFF (Wang et al., 2024a) demonstrate the potential to enhance existing forecasters using learned timestamp encodings, our approach primarily focuses on integrating paired metadata with temporal information for improved forecasting accuracy. Our architecture uses separate embeddings and cross-attention layers for temporal and non-temporal data. In Table 13, we present results comparing the effects of using only temporal information (commonly available in time series) versus utilizing the full contextual information, which includes both timestamps and associated metadata.

| MODEL | | PATCHTST | | | | ITRANSFORMER | | | |
|---|---|---|---|---|---|---|---|---|---|
| CONTEXT | | TEMPORAL | | FULL | | TEMPORAL | | FULL | |
| DATASET | T | MAE | MSE | MAE | MSE | MAE | MSE | MAE | MSE |
| AIR QUALITY | 48 | 0.537 | 0.702 | **0.524** | **0.674** | 0.559 | 0.730 | **0.540** | **0.696** |
| | 96 | 0.587 | 0.829 | **0.572** | **0.802** | 0.607 | 0.858 | **0.591** | **0.813** |
| ELECTRICITY | 48 | 0.033 | 0.044 | **0.029** | **0.036** | 0.031 | 0.040 | **0.028** | **0.035** |
| | 96 | 0.029 | 0.034 | **0.024** | **0.027** | 0.027 | 0.032 | **0.024** | **0.028** |
| TRAFFIC | 48 | 0.912 | 2.933 | **0.865** | **2.922** | 0.882 | **2.856** | **0.848** | 2.868 |
| | 96 | 0.901 | 2.819 | **0.845** | **2.767** | 0.882 | 2.799 | **0.830** | **2.766** |
| RETAIL | 48 | 0.119 | 0.232 | **0.115** | **0.228** | 0.127 | 0.260 | **0.124** | **0.257** |
| | 96 | 0.133 | 0.276 | **0.128** | **0.265** | 0.144 | **0.307** | **0.143** | 0.310 |
| BITCOIN | 48 | 0.833 | 1.204 | **0.821** | **1.192** | 0.831 | 1.244 | **0.810** | **1.153** |
| | 96 | 0.949 | 1.538 | **0.948** | **1.537** | 0.952 | **1.538** | **0.951** | 1.547 |

Table 13: **Temporal Information Vs. Full Context.** We compare the performance of the context-aware PatchTST and iTransformer models for time series forecasting, using both temporal information alone and the full context, which includes timestamps and metadata, with a fixed lookback length of $L = 96$ and prediction lengths of $T \in \{48, 96\}$. Incorporating the full contextual information, including metadata, leads to significant performance improvements over using only temporal data across most datasets.

# F  ADDITIONAL QUALITATIVE PLOTS

In this section, we highlight the top three examples of both improvement and degradation in forecast quality after incorporating ContextFormer across various base architectures and datasets. This provides an unbiased comparison between context-aware and context-agnostic forecasts. Note that all plots in this section correspond to a forecasting horizon of 96 steps.

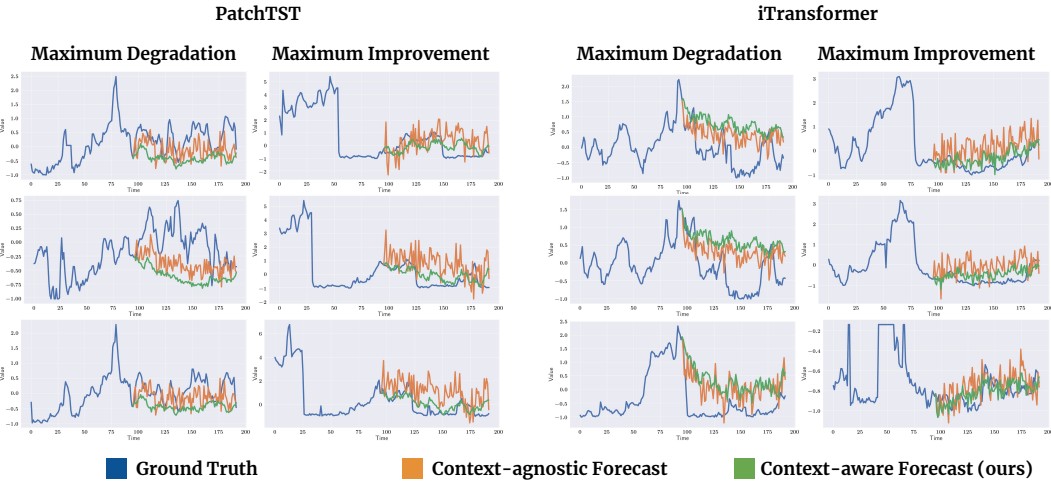

Figure 8: **Qualitative Plots for Air Quality Dataset.** The plots showcase the top three examples with the highest degradation and the improvement in MSE for the ContextFormer forecasts compared to the baseline.

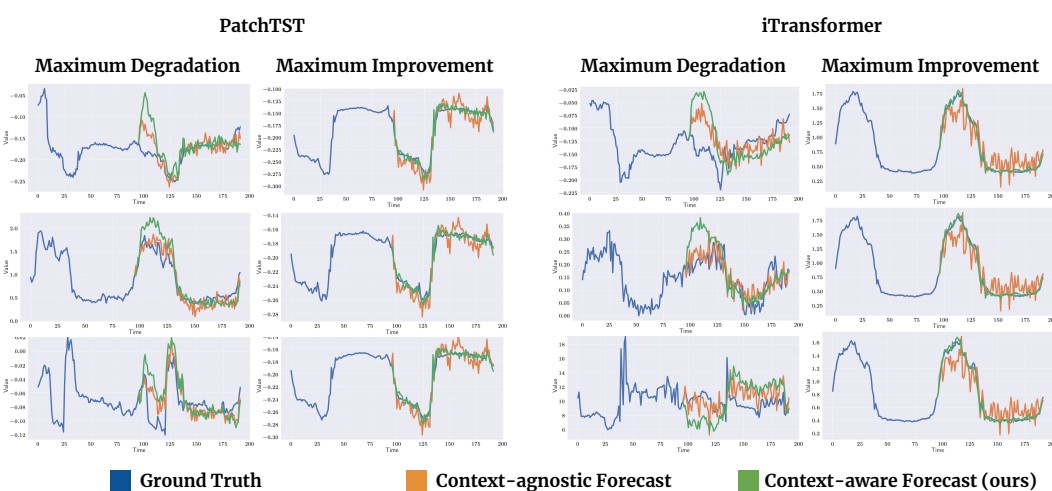

Figure 9: **Qualitative Plots for Electricity Dataset.** The plots showcase the top three examples with the highest degradation and the improvement in MSE for the ContextFormer forecasts compared to the baseline.

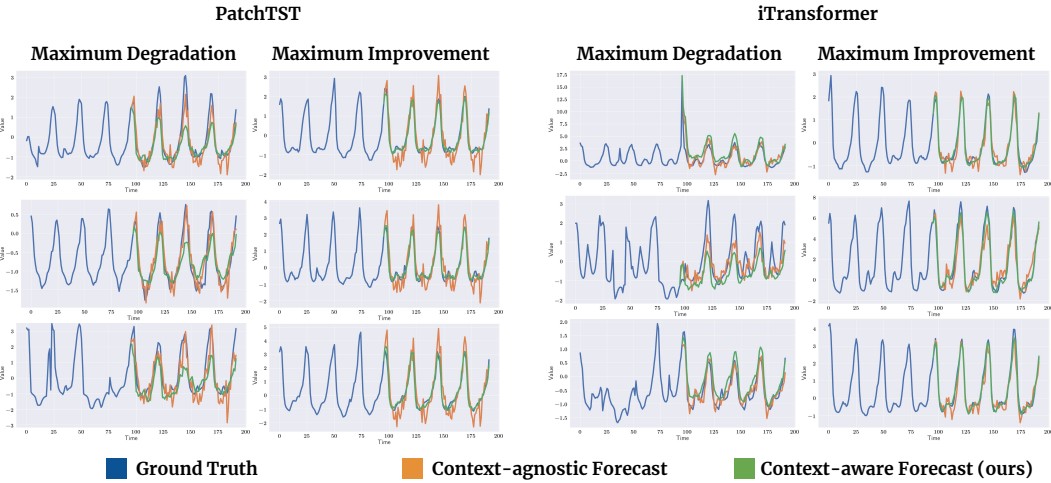

Figure 10: **Qualitative Plots for Traffic Dataset.** The plots showcase the top three examples with the highest degradation and the improvement in MSE for the ContextFormer forecasts compared to the baseline.

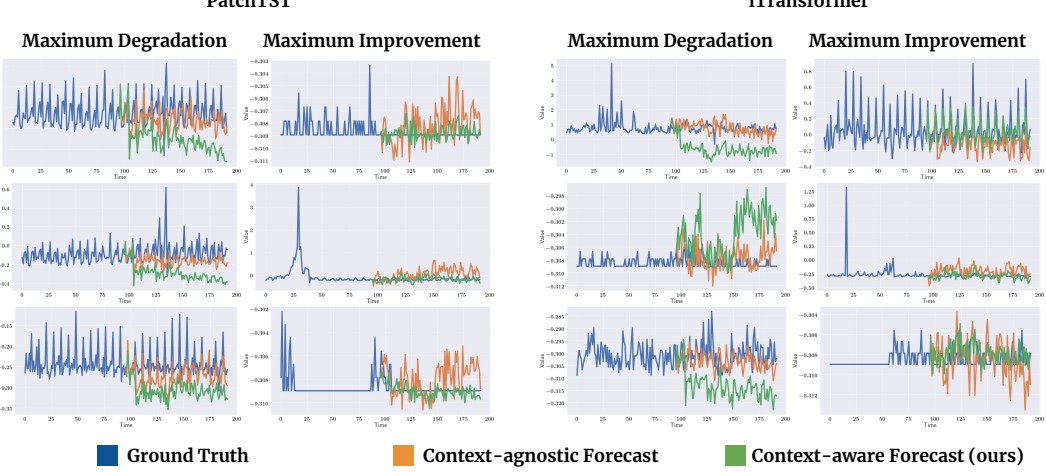

Figure 11: **Qualitative Plots for Retail Dataset.** The plots showcase the top three examples with the highest degradation and the improvement in MSE for the ContextFormer forecasts compared to the baseline.

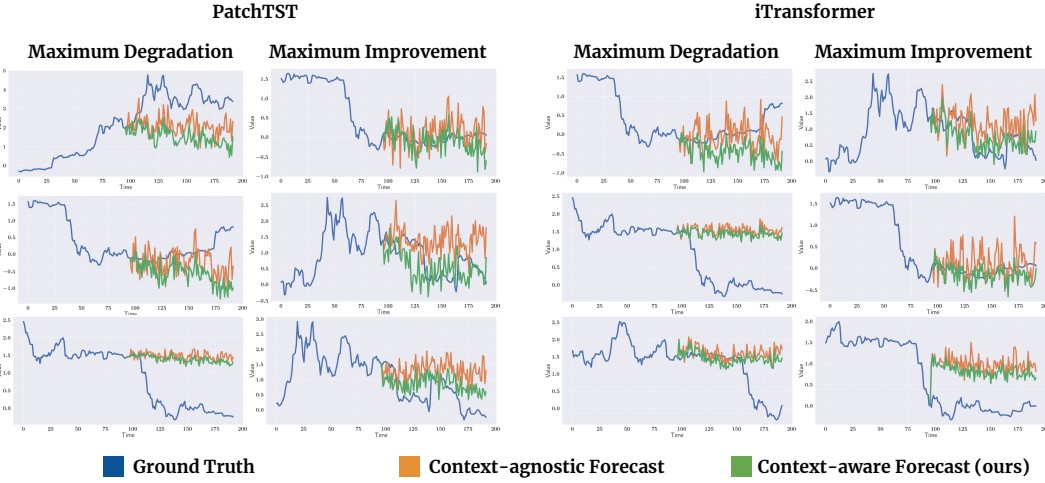

Figure 12: **Qualitative Plots for Bitcoin Dataset.** The plots showcase the top three examples with the highest degradation and the improvement in MSE for the ContextFormer forecasts compared to the baseline.

