# OpenReview forum: "Context Matters: Leveraging Contextual Features for Time Series Forecasting"
_ICLR.cc/2025/Conference — Submitted to ICLR 2025_

### Official Review · Reviewer_ML1Z · 2024-10-16

**Soundness:** 2
**Presentation:** 2
**Contribution:** 2
**Rating:** 5
**Confidence:** 4

**Summary:**

This paper introduces ContextFormer, a method designed to incorporate multimodal contextual information into existing time series forecasting models. Traditional models typically rely solely on historical data, neglecting external influences such as news, weather, and market trends that can significantly impact predictions. ContextFormer effectively addresses this limitation by integrating diverse types of metadata—including categorical, continuous, time-varying, and textual information—using cross-attention mechanisms. The experiments demonstrate that ContextFormer can enhance forecasting performance by up to 30% compared to state-of-the-art models across various datasets in fields such as energy, traffic, environment, and finance.

**Strengths:**

1. As a plug-in solution, ContextFormer can be integrated with various predictive backbones for a diverse array of applications.
2. The two-stage fine-tuning methodology guarantees that the lower bound of ContextFormer is at least equivalent to that of context-agnostic approaches.
3. The extensive experiments look hopeful.

**Weaknesses:**

1. **Inadequate Presentation.** The organization of the paper lacks rationality. Excessive emphasis is placed on the importance of external context, which is well-known. The methods section is overly succinct. The descriptions of Metadata Embedding and Temporal Embedding are overly concise, leaving the application dimensions and the output shape ambiguous. Furthermore, the embedded interaction in the Information Fusion component is similarly unclear, too. Its applicability to both a single embedding for one variable (iTransformer and PatchTS) and a single embedding for one timestamp (Informer and Autoformer) across two distinct architectures remains uncertain. In summary, the workflow of the algorithm is perplexing.

2. **Limited Innovativeness.** Although the author presents a case in the introduction regarding the influence of news on stock prices, the treatment of this unstructured external text information is only mentioned in Appendix C.7 cursory and lacks corresponding experimental results on BITCOIN-NEWS. The models and data discussed throughout the paper primarily rely on structured auxiliary information, despite incorporating both continuous and categorical variables. The prior research has focused on integrating structured external information to enhance prediction accuracy, including exogenous variables [1] and timestamps [2]. The paper's core contribution remains ambiguous. Moreover, The three challenges outlined in the introduction are unpersuasive.

3. **Unrigorous Experiments.** As stated in Weaknesses 2, there has been some prior work aimed at integrating structured external information to enhance prediction accuracy. This paper should be compared with these frameworks that utilize external information, rather than solely with models that lack auxiliary information.

**References**

[1] 2024, TimeXer: Empowering Transformers for Time Series Forecasting with Exogenous Variables

[2] 2024, Rethinking the Power of Timestamps for Robust Time Series Forecasting: A Global-Local Fusion Perspective

**Questions:**

The authors can refer to the weakness listed above.

---

> ### Author Response · Authors · 2024-11-22
> **Response To Reviewer ML1Z**
>
> We thank the reviewer for their thoughtful review and for recognizing the simplicity, versatility, and effectiveness of the proposed ContextFormer in integrating contextual information to enhance forecasting performance across diverse applications.
>
> # Addressing the weaknesses indicated by the reviewer.
>
> ## Response to Weakness 1
>
> **On the lack of methodology details.** The dimensions and processing for both the metadata and temporal embeddings are provided in **Appendix D2.2 (Page 18)** and **Table 9**, while details for the base models are given in **Appendix D2.1**.  We have updated some details for the metadata embedding to clarify a doubt raised by reviewer ygXG. Also, we have added references to the appendix sections containing the details for the architecture and implementation in **Page 6**. The metadata embeddings are obtained for each timestamp in our models. The complete source code will be released after publication.
>
> ## Response to Weakness 2
>
> **On the lack of Innovativeness.**
>
> While there has been prior research on utilizing structured external information for context-aware models, we propose a novel approach to building such forecasting models from the existing architectures Below, we list the three main contributions of our approach.
>
>
> **a) Framework for enhancing pre-existing forecasters through contextual information**-  Most of the SOTA forecasting models rely solely on historical data; these models can significantly benefit from our ContextFormer fine-tuning. The technique could be extended to adapt large time-series foundational models like TimesFM and  Chronos on dataset-specific contextual information.
>
> Currently, the most common strategy to adapt pre-trained models like TimesFM to contextual information is through fitting exogenous linear models onto the covariates, but these models are limited in their ability to capture complex correlations and also depend on the availability of future covariate values, making them unsuitable for many real-world problems.
>
> **b) Ability to handle diverse metadata types and base architecture-** Our strategy can adapt and enhance any pre-existing forecasting model, regardless of the architecture, by utilizing any available contextual information—be it continuous or discrete, time-varying or time-invariant.
>
> **c) Plug-and-play fine-tuning approach-** Our setup setup allows forecasters to incorporate domain-specific contextual information alongside the time-series features learned during pre-training, providing a guarantee for the context-aware model to be at least as good as the context-agnostic one, without having to modify the entire model.
>
> In the future, our technique could be extended to adapt large time-series foundational models like TimesFM and  Chronos for domain-specific tasks in a parameter-efficient manner without having to modify the entire model.
>
> **On the BITCOIN-News dataset.** We conducted separate experiments to account for the unique nature of the context, which consisted of textual data. This dataset was included in our experiments, and as shown in **Table 4 (Page 10)**, incorporating context into our model led to an average improvement of **11.5%** in MAE, demonstrating the potential of our approach. We have added further information on the Bitcoin-News dataset in Appendix C.8.
>
> ## Response to Weakness 3
>
> **On the rigorousness of experiments.** As per the reviewers’ suggestions, we have included a comparison of our models against TiDE and TimeXer in **Table 2 (Page 8). The ContextFormer-enhanced models outperform existing context-aware baselines like TimeXer & TiDE on the majority of the datasets. ContextFormer achieves up to 26% improvement over TimeXer (with an average of 6.6% across all settings) and up to 8.6% improvement over TiDE (averaging 4.1% across all settings) in terms of MSE.** The comparison is fair as baselines and ContextFormer-enhanced models take the same input (history + metadata). Additionally, we note that TimeXer outperforms TimesNet on forecasting tasks for real-world datasets. Note that the **performance of our context-aware models is limited by the constraints of base architectures**; thus, given that we will have more effective base models in the future, our technique could outperform any existing models that natively support covariates.
>
> **On GLAFF.** We thank you for bringing the GLAFF paper to our notice; we have cited the paper in our discussion on the temporal information given in Appendix E.4. The paper demonstrates the potential of enhancing existing forecasters using learned timestamp encodings. Meanwhile, our approach primarily focuses on integrating paired contextual metadata, along with such temporal information, to improve the forecasting accuracy.

---

> > ### Comment · Reviewer_ML1Z · 2024-11-22
> >
> > The author has basically resolved my concerns, and I will improve my score.

---

> > > ### Author Response · Authors · 2024-11-24
> > > **Thank you for increasing the score**
> > >
> > > Dear Reviewer ML1Z,
> > >
> > > We are glad that our rebuttal has addressed your concerns. We sincerely appreciate your thoughtful review and your decision to increase the score.

---

### Official Review · Reviewer_hCqX · 2024-10-29

**Soundness:** 3
**Presentation:** 3
**Contribution:** 2
**Rating:** 5
**Confidence:** 3

**Summary:**

This article proposes ContextFormer, a novel plug-and-play framework that utilizes cross-attention blocks to integrate multimodal metadata, including categorical, continuous, time-varying contexts, and even textual information, into existing any context-agnostic base forecasters. The author selects two SOTA models of PatchTST and iTransformer as the base forecasters and validates ContextFormer framework on seven real-world datasets spanning energy, traffic, environmental, and financial domains. Experimental results confirm the effectiveness of the framework.

**Strengths:**

1. The starting point of this article is good. Exogenous contextual features can indeed serve as key auxiliary information to influence time series forecasting.
2. The paper is mostly well-presented and easy to follow.

**Weaknesses:**

1. Introduction part summarizes three challenges in incorporating metadata into forecasting models, but this article does not provide a detailed explanation of why ContextFormer can solve these problems. For example, the first challenge mentions the importance of aligning time series history and multimodal metadata. However, there is a lack of evidence on how ContextFormer ensures that the two have been aligned, the alignment effect, and the impact of aligned representation on forecasting results. Although the effectiveness of ContextFormer is reflected in the final MSE/MAE metrics, I mainly focus on these intermediate results and suggest the author to supplement these explanations.

2. In related works part, the author lists some methods for forecasting with covariates. As far as I know, it also includes methods such as TFT, TSMixer, and TimeXer, etc., but this article does not compare with these methods. It is suggested that the author supplements these comparison experiments.
Some references:
TFT: https://doi.org/10.1016/j.ijforecast.2021.03.012,
TSMixer: https://arxiv.org/pdf/2303.06053,
TimeXer: https://arxiv.org/pdf/2402.19072.

3. In Table 2 caption, it is mentioned that "The best results for each base model in each row are highlighted in bold.". However, the results of the Retail dataset (horizon=96, MSE metric) do not match this statement. It is recommended to revise the wording.

4. Doubt about lines 491-492:
    1) The explicit motivation for comparing with Chronos and whether the comparison is fair?
    2) Are the results in Table 2 sufficient to support this conclusion? I agree with the example provided by the author (486~487), but for other datasets, I have the following questions: (a) Is it still necessary to compare the context-aware model with Chronos for the two datasets of Air quality and Electricity, as the context-agnostic model is already better than Chronos? (b) For the Retail and Bitcoin datasets, Chronos performs the best. I hope the author can provide detailed explanations to alleviate my concerns.

**Questions:**

Refer to "Strengths And Weaknesses".

---

> ### Author Response · Authors · 2024-11-22
> **Response To Reviewer hCqX**
>
> We sincerely thank the reviewer for their thoughtful and detailed review, as well as for their kind appreciation of the paper's motivation and presentation.
>
> # Addressing the weaknesses indicated by the reviewer.
>
> ## Response to Weakness 1
>
> **On metadata alignment.** During ContextFormer fine-tuning, the metadata is passed through the model along with the corresponding time series and time stamp. We necessitate, by design, that each metadata point must align with a data sample and, therefore, be aligned with a timestamp. This alignment is trivial for time-invariant metadata, which remains constant for all data points within a given series. In scenarios like the Bitcoin-News dataset, where the data and metadata have different granularities, we align the metadata with the data by either subsampling or supersampling. Specifically, for the given example with hourly closing values and daily news articles, the articles are repeated across the 24 data points corresponding to each day.
>
> ## Response to Weakness 2
>
> **On limited relevant literature.** Thank you for your review; we have updated the relevant literature section to include more models that take into account covariates, such as **TFT, TimeMixer, TiDE and TimeXer.**
>
> **On the comparison with SOTA context-aware forecasters.** As per the reviewers’ suggestions, we have included a comparison of our models against TiDE and TimeXer in **Table 2 (Page 8). The ContextFormer-enhanced models outperform existing context-aware baselines like TimeXer & TiDE on the majority of the datasets. ContextFormer achieves up to 26% improvement over TimeXer (with an average of 6.6% across all settings) and up to 8.6% improvement over TiDE (averaging 4.1% across all settings) in terms of MSE.** The comparison is fair as baselines and ContextFormer-enhanced models take the same input (history + metadata). Additionally, we note that TimeXer outperforms TimesNet on forecasting tasks for real-world datasets. Note that the **performance of our context-aware models is limited by the constraints of base architectures**; thus, given that we will have more effective base models in the future, our technique could outperform any existing models that natively support covariates.
>
> ## Response to Weakness 3
>
> **Clarification** **in Table 2 caption.** We have updated the caption to clarify that the best results for each base architecture (between the original and the ContextFormer-enhanced model) are highlighted in **bold**. The best overall result across all architectures for each row is now **underlined** for a clearer comparison.
>
> ## Response to Weakness 4
>
> **On the comparison with Chronos.** Our comparison of Chronos with the ContextFormer-enhanced model aimed to demonstrate that smaller models, with context-aware fine-tuning, can rival or outperform massive pre-trained forecasters. We understand that the results were unable to highlight this. In a few datasets where our base models outperformed Chronos, the additional comparison with ContextFormer enhanced models did not yield any significant insight. For datasets like bitcoin, where Chronos surpasses the base models by a huge margin, even our enhancements could not cover such a huge gap. With the availability of more effective base-models in future, we expect the ContextFormer-enhanced models to be able to uniformly outperform such foundational models.
>
> We have shifted the comparison with foundational models to **Appendix E.2 (Page 20)**. We believe that using context-aware models like TimeXer and TiDE as a baseline provides a clearer demonstration of our approach's qualities.

---

> > ### Author Response · Authors · 2024-11-24
> > **Requesting Feedback on the Rebuttal**
> >
> > Dear Reviewer hCqX,
> >
> > We hope our rebuttal and the changes made to the draft have addressed the questions and weaknesses pointed out in your review. We are happy to address further questions or concerns about our contributions during the discussion phase. If our clarifications and the newly added results meet your expectations, we kindly request you to consider revising the score.

---

> ### Author Response · Authors · 2024-12-02
> **Requesting Feedback on the Rebuttal**
>
> Dear Reviewer hCqX,
>
> We hope our rebuttal and the changes made to the draft have addressed the concerns raised in your review. As the discussion period concludes today, we kindly request you to let us know if we need to provide any more clarifications. If our clarifications and the newly added results align with your expectations, we kindly request your consideration for a score revision.

---

> > ### Comment · Reviewer_hCqX · 2024-12-03
> >
> > Thanks for your responses. While some of our concerns have been adequately clarified, I still find certain explanations and justifications are insufficient, as summarized below:
> >
> > 1. **More evidence is needed to prove that ContextFormer can solve the three problems mentioned in the Introduction part.** Aligning time series data with multimodal metadata is indeed a significant challenge. Although the authors provide an explanation for aligning time series data and metadata through timestamps, **this is only during the data preprocessing stage**. **What I am more interested in is whether the model can indeed achieve alignment during the ContextFormer fine-tuning**. For the other two challenges, the authors seem to have not provided a more detailed explanation of how ContextFormer can solve them.
> >
> > 2. From Table 2, the forecasting performance of ContextFormer seems to be related to the choice of base model. However, this article only provides the experimental results of two base models (PatchTST and iTransformer), which is **clearly insufficient to demonstrate the advantages of ContextFormer**.
> >
> > Based on these concerns, I would like to maintain my origin score.

---

> > > ### Author Response · Authors · 2024-12-03
> > >
> > > Thank you for your response. We would be grateful if you could go through the following reply, which we hope addresses your concerns.
> > >
> > > ## Response to Concern 1
> > >
> > > For the three challenges mentioned in the introduction:
> > >
> > > i) **The cross-attention mechanism between the time series and metadata embeddings during ContextFormer fine-tuning ensures a robust alignment** between the forecasted quantity and the associated contextual information, enabling the model to effectively integrate and leverage relevant metadata for improved predictions.
> > >
> > > ii) As noted in the second challenge, **the lack of uniformity in metadata has hindered the development of a context-aware foundational model.** The existing adaptation approaches, such as fitting exogenous linear models to covariates, are limited in capturing complex correlations and rely on the availability of future covariate values, rendering them unsuitable for most real-world scenarios. Our approach addresses this exact issue through ContextFormer fine-tuning to adapt pre-trained models to new datasets while effectively handling complex nonlinear relationships without requiring future metadata values.
> > >
> > > iii) While numerous context-aware models exist, many overlook the diversity of metadata types, such as continuous, discrete, time-varying, or time-invariant data, thus often neglecting the improvement that could have been obtained from simple discrete contextual information. Our methodology addresses this limitation by **processing continuous and discrete metadata separately** using dedicated encoders optimized for their respective needs. The outputs are then integrated and passed through the ContextFormer pipeline for comprehensive modeling.
> > >
> > > ## Response to Concern 2
> > >
> > > We selected PatchTST and iTransformer as the context-agnostic base models because of their **state-of-the-art forecasting performance** and **contrasting architectural designs**, thus allowing us to effectively demonstrate ContextFormer’s potential to **generalize and adapt across varying base-model paradigms**. These model choices were also influenced by their recognition among the widely used benchmarks, as seen in the popular **THUML Time Series Library** on GitHub (https://github.com/thuml/Time-Series-Library#leaderboard-for-time-series-analysis). By improving performance on these two contrasting models, ContextFormer demonstrates its potential for improving the existing models irrespective of the base architecture, hence paving the way for any future context-agnostic baselines. We hope this clarifies our rationale for the choice of these two models and addresses your concerns.
> > >
> > >
> > > Thank you again for your thoughtful comments. As the reviewer response period ends today AoE, we urge you to kindly reconsider your score.

---

### Official Review · Reviewer_wzvi · 2024-11-03

**Soundness:** 3
**Presentation:** 3
**Contribution:** 1
**Rating:** 5
**Confidence:** 5

**Summary:**

This paper presents a framework designed to incorporate exogenous variables into time series forecasting.

**Strengths:**

1. The presentation is clear.
2. The problem studied is interesting.

**Weaknesses:**

1. The contribution is limited.
2. There is a lack of comparison with SOTA models.

**Questions:**

The presentation and problem setup are clear and easy to follow. However, I have two major concerns regarding the theoretical analysis of the benefits of adding exogenous variables and the overall technical contribution of the paper.
1. By definition, mutual information is non-negative. Even in the worst-case scenario where A and C are completely independent, the increase in mutual information is 0. A higher mutual information value does not necessarily translate to higher accuracy or better performance. Noisy or even misleading data can also increase mutual information as long as some degree of dependency exists. Could the authors elaborate on this issue?
2. Adding exogenous variables to time series forecasting models is not a novel concept. It is a natural extension of general time series forecasting models and is a dominant approach for work in some domains, such as financial market predictions. For example, older transformer-based models like Autoformer can be adapted to include exogenous variables and are often used as baselines in other papers. Moreover, newer transformer-based models, such as Timexer, also support this approach natively. There are also attempts to use large language models (LLMs) to incorporate exogenous market information for stock movement prediction, such as Plutos. Therefore, I believe the overall contribution of the work is somewhat weak, as it does not introduce a new concept.
3. It is expected that the authors should at least compare the proposed framework with other models that can accept the same input. Otherwise, it is difficult to justify the performance of the proposed framework.

---

> ### Author Response · Authors · 2024-11-22
> **Response To Reviewer wzvi**
>
> We thank the reviewer for taking the time to provide a thoughtful and thorough review of the paper. We are glad that the reviewer finds the problem interesting and appreciates the clarity of the presentation.
>
> # Addressing the weaknesses indicated by the reviewer.
>
> ## Response to Weakness 1
>
> **Key Contributions:**
>
> **a) Framework for enhancing pre-existing forecasters through Contextual information**-  Most of the SOTA forecasting models rely solely on historical data; these models can significantly benefit from our ContextFormer fine-tuning. The technique could be extended to adapt large time-series foundational models like TimesFM and  Chronos on dataset-specific contextual information.
>
> Currently, the most common strategy to adapt pre-trained models like TimesFM to contextual information is through fitting exogenous linear models onto the covariates, but these models are limited in their ability to capture complex correlations and also depend on the availability of future covariate values, making them unsuitable for many real-world problems.
>
> **b) Ability to handle diverse metadata types and base architecture-** Our strategy can adapt and enhance any pre-existing forecasting model, regardless of the architecture, by utilizing any available contextual information—be it continuous or discrete, time-varying or time-invariant.
>
> **c) Plug-and-play fine-tuning approach-** Our setup setup allows forecasters to incorporate domain-specific contextual information alongside the time-series features learned during pre-training, providing a guarantee for the context-aware model to be at least as good as the context-agnostic one, without having to modify the entire model.
>
> ## Response to Weakness 2
>
> **The ContextFormer-enhanced models outperform existing context-aware baselines like TimeXer & TiDE on the majority of the datasets including ETT.** As per the reviewers’ suggestions, we have included a comparison of our models against TiDE and TimeXer, which are SOTA context-aware forecasting models, in **Table 2 (Page 8). ContextFormer achieves up to 26% improvement over TimeXer (with an average of 6.6% across all settings) and up to 8.6% improvement over TiDE (averaging 4.1% across all settings) in terms of MSE.** The comparison is fair as baselines and ContextFormer-enhanced models take the same input (history + metadata). Additionally, we note that TimeXer outperforms TimesNet on forecasting tasks for real-world datasets. Note that the performance of our context-aware models is limited by the constraints of base architectures; thus, given that we will have more effective base models in the future, our technique could outperform any existing models that natively support covariates.
>
> # Addressing the questions asked by the reviewer.
>
> ## Response to Question 1
>
> **Maximizing mutual information minimizes the MSE loss.**  [1] has proven it for retrieval-based forecasting, under the assumption of gaussian noise between the ground truth and forecast.(, We extend this equivalence for context-aware forecasting and provide a discussion in Appendix A. Because mutual information is inherently non-negative, it does not decrease even in the presence of noisy metadata. In the worst-case scenario, where the model fails to identify any meaningful correlation between the metadata and the time-series data, the performance remains unchanged. This theoretical analysis highlights an essential insight. **For any given base model, a context-aware setup consistently outperforms or at least matches the performance of a context-agnostic one, regardless of the specific model or dataset.**
>
> ## Response to Question 2
>
> We acknowledge that incorporating useful context for forecasting is not a novel problem. However, our primary focus is on developing a model and training strategy that enables **seamless addition of context to any state-of-the-art forecaster.** This design ensures that as context-agnostic forecasters improve, the performance of the context-enhanced model (ContextFormer) will also improve proportionally. In response to reviewers' suggestions, we have included comparisons with existing SOTA forecasters that utilize context, such as **TiDE** and **TimeXer**, to further validate our approach. We have enhanced two base architectures, PatchTST and iTransformer, both of which already outperform AutoFormer. We have also added a citation for Ploutos in the related works section.
>
> ## Response to Question 3
>
> As per the reviewers’ suggestions, we have added baseline comparisons, which we mentioned in our response to Weakness 2.
>
> [1] Retrieval Based Time Series Forecasting- https://[arxiv.org/abs/2209.13525](http://arxiv.org/abs/2209.13525)

---

> > ### Author Response · Authors · 2024-11-24
> > **Requesting Feedback on the Rebuttal**
> >
> > Dear Reviewer wzvi,
> >
> > We hope our rebuttal and the changes made to the draft have addressed the questions and weaknesses pointed out in your review. We are happy to address further questions or concerns about our contributions during the discussion phase. If our clarifications and the newly added results meet your expectations, we kindly request you to consider revising the score.

---

> > ### Author Response · Authors · 2024-12-02
> > **Requesting Feedback on the Rebuttal**
> >
> > Dear Reviewer wzvi,
> >
> > We hope our rebuttal and the changes made to the draft have addressed the concerns raised in your review. As the discussion period concludes today, we kindly request you to let us know if we need to provide any more clarifications. If our clarifications and the newly added results align with your expectations, we kindly request your consideration for a score revision.

---

### Official Review · Reviewer_ygXG · 2024-11-04

**Soundness:** 3
**Presentation:** 2
**Contribution:** 1
**Rating:** 5
**Confidence:** 4

**Summary:**

This paper presents a work on aggregating contextual information into time series forecasting. Specifically, the authors propose to use a universal context encoder to encode the contextual information as embedding and boost the time series forecastor with the encoding. Experimental results suggest that the contextual information boost the performance.

**Strengths:**

The writing is easy to follow

**Weaknesses:**

1. The contribution is limited. This paper mainly discuss about contextual information in time series forecasting. However, it is well known that the contextual information is helpful for time series. What really matters is how to make use of it. In this paper, the authors list three reasons about why it is hard to make use of the information, such as multi modality and non-uniformity across domains. However, when really talking about how to make use of the context, the authors just simply concatenate the category variables with continuous variables. This design is unrelevant with the challenges mentioned. What if some out-of-domain dataset contain some variables that are unseen on the training data? Also, the design is well-justified. Why not align the metadata as one modality (like text) and then convert it to embeddings with one shared encoder? Do we need to align the timely metadata (like news with timestamp) with the speific timestamp before forwarding them all into the cross attention?

2. The theoretic analysis is useless but takes 1.5 pages. All the theoretic results are trivial corollary of conclusions from introductory undergraduate-level text book of information theory and machine learning. I suggest that the authors are just trying to decorate their papers with some theoretic analysis

3. Experimental results are not comprehensive enough. We prefer experiments on more datasets (like ETT) and baselines (like TimesNet).

4. The authors list 4 reasons why they prefer fine-tuning in Sec 5.2. But there is lack of empirical supports. I did not see training curves that reflect the unstable training of all-from-scratch.

5. The bitcoin-news dataset description is too short. This could be the most insightful part about how to collect contextual data. But there is no details. How and where did you get the data? Are the data filtered? Are the collection based on time or keywords? Some many details are missing.

**Questions:**

See the weakness

---

> ### Author Response · Authors · 2024-11-22
> **Response To Reviewer ygXG (Part 1)**
>
> We thank the reviewer for taking the time to provide a thoughtful and thorough review of the paper and for recognizing the clarity of this writeup.
>
> # Addressing the weaknesses indicated by the reviewer.
>
> ## Response to Weakness 1
>
> The major concerns raised by the reviewer and the corresponding replies are given below-
>
> ### **1.1 On limited contributions.**
>
> Below, we list the three main contributions of our approach.
>
>
> **a) Framework for enhancing pre-existing forecasters through contextual information**-  Most of the SOTA forecasting models rely solely on historical data; these models can significantly benefit from our ContextFormer fine-tuning. The technique could be extended to adapt large time-series foundational models like TimesFM and  Chronos on dataset-specific contextual information.
>
> Currently, the most common strategy to adapt pre-trained models like TimesFM to contextual information is through fitting exogenous linear models onto the covariates, but these models are limited in their ability to capture complex correlations and also depend on the availability of future covariate values, making them unsuitable for many real-world problems.
>
> **b) Ability to handle diverse metadata types and base architecture-** Our strategy can adapt and enhance any pre-existing forecasting model, regardless of the architecture, by utilizing any available contextual information—be it continuous or discrete, time-varying or time-invariant.
>
> **c) Plug-and-play fine-tuning approach-** Our setup setup allows forecasters to incorporate domain-specific contextual information alongside the time-series features learned during pre-training, providing a guarantee for the context-aware model to be at least as good as the context-agnostic one, without having to modify the entire model.
>
>
> ### **1.2 On the relevance of design to the challenges.**
> To effectively handle the distinct metadata types, our architectural design **first embeds the categorical and continuous variables separately before concatenating them and further processing them through the transformer encoder** as detailed in Appendix D.2.2. We have revised Section 5.1 to clarify this process and corrected the text that previously implied direct concatenation of metadata features prior to embedding.
>
> ### **1.3 On handling metadata that was never seen during training**.
>
> The input metadata variables need to be defined for context-aware models. Consider the air quality example. Such models cannot be trained with humidity and temperature metadata, and accept wind speed as additional input during test time. We note that **this is a well-known drawback for all the state-of-the-art context-aware forecasters**, such as TiDE, TimeXer, and not specific to our approach.
>
> ### **1.4 On converting metadata to text can result in significant information loss and add additional computational overhead.**
>
> While converting metadata into text and utilizing the corresponding text embeddings seems lucrative, it has two potential disadvantages. First, it can result in **significant information loss**, as the conversion process may fail to capture the full granularity or structure of the original metadata, leading to less effective representations. Second, it introduces **additional computational overhead**, as generating and processing text embeddings can be resource-intensive, especially for large-scale datasets or real-time applications. Moreover, converting the time-varying metadata sequence through textual embedding may also cause data metadata alignment details to be lost.
>
> ### **1.5 On metadata alignment.**
>
> During ContextFormer fine-tuning, the metadata is passed through the model along with the corresponding time series and time stamp. **We necessitate, by design, that each metadata point must align with a data sample and, therefore, be aligned with a timestamp**. This alignment is trivial for time-invariant metadata, which remains constant for all data points within a given series. In scenarios like the Bitcoin-News dataset, where the data and metadata have different granularities, we align the metadata with the data by either subsampling or supersampling. Specifically, for the given example with hourly closing values and daily news articles, the articles are repeated across the 24 data points corresponding to each day.

---

> ### Author Response · Authors · 2024-11-22
> **Response To Reviewer ygXG (Part 2)**
>
> ## Response to Weakness 2
>
> The theoretical analysis provides a critical insight that **for the same base model, a context-aware setup would be a better forecaster than a context-agnostic one**, irrespective of the model or dataset. We have made minor changes to Section 4.1 to make it more concise following the reviews.
>
> Furthermore, **our design choice of using zero-initialized weights for ContextFormer’s fine-tuning phase is inspired by the theoretical justification in Section 4.2**, which provides groundwork for definitive performance improvement while adding external information (metadata) in linear settings.
>
> ## Response to Weakness 3
>
> **The ContextFormer-enhanced models outperform existing context-aware baselines like TimeXer & TiDE on most datasets including ETT.** We have included a comparison of our models against TiDE and TimeXer, which are SOTA context-aware forecasting models, in **Table 2 (Page 8). ContextFormer achieves up to 26% improvement over TimeXer (with an average of 6.6% across all settings) and up to 8.6% improvement over TiDE (averaging 4.1% across all settings) in terms of MSE.** The comparison is fair as baselines and ContextFormer-enhanced models take the same input (history + metadata). Additionally, we note that TimeXer outperforms TimesNet on forecasting tasks for real-world datasets.
>
> ## Response to Weakness 4
>
> **Empirical Support for fine-tuning has been provided in Table 3 and Appendix E.2.** The fine-tuned models always outperform the base models, unlike the models trained from scratch. The empirical results are provided in **Table 3 (Page 9)** of the main text and further elaborated in **Appendix E.2**. A comparison of **training curves for full training and fine-tuning** has been added in the same section. The results on the Bitcoin dataset for a forecast length of 96 demonstrate that fine-tuning allows convergence to a lower validation loss, while full training causes the model's validation loss to diverge much earlier than during fine-tuning, as proposed in the paper.
>
> In the majority of experiments, fine-tuning outperforms training from scratch. Unstable training could be one of the potential explanations for this phenomenon. Another plausible explanation could be the difficulty of the base architecture in simultaneously learning historical and covariate relationships. These challenges may vary depending on the dataset and base models. Therefore, we propose ContextFormer fine-tuning, which consistently delivers better performance than the base model.
>
> ## Response to Weakness 5
>
> **We have updated the description of the Bitcoin-news dataset in Appendix C.8.** The articles from January 1st, 2022, to February 17th, 2024, were sourced using the **Alpaca Historical News API** (doc- https://docs.alpaca.markets/docs/historical-news-data), with each metadata instance consisting of all news articles and headlines tagged with BTCUSD} for a given day. These textual instances were directly processed (without additional filtering) using the **OpenAI Embeddings model `text-embedding-1-small'**, producing 1536-dimensional embeddings for each day’s news. To ensure causality, the hourly Bitcoin closing prices for a given day were aligned with the previous day’s news embeddings. These embeddings served as time-varying metadata, remaining constant within a day but varying daily.We will be happy to answer any more questions, the complete source code will be released after publication.

---

> > ### Author Response · Authors · 2024-11-24
> > **Requesting Feedback on the Rebuttal**
> >
> > Dear Reviewer ygXG,
> >
> > We hope our rebuttal and the changes made to the draft have addressed the questions and weaknesses pointed out in your review. We are happy to address further questions or concerns about our contributions during the discussion phase. If our clarifications and the newly added results meet your expectations, we kindly request you to consider revising the score.

---

> ### Author Response · Authors · 2024-12-02
> **Requesting Feedback on the Rebuttal**
>
> Dear Reviewer ygXG,
>
> We hope our rebuttal and the changes made to the draft have addressed the concerns raised in your review. As the discussion period concludes today, we kindly request you to let us know if we need to provide any more clarifications. If our clarifications and the newly added results align with your expectations, we kindly request your consideration for a score revision.

---

### Author Response · Authors · 2024-11-27
**Summary of Revisions**

We thank all the reviewers for their time and very thoughtful comments. We would like to put forward the key contribution of this paper and summarize the revisions made during the rebuttal period.

## **Key Contributions.**

1) **A framework for enhancing pre-existing forecasters through contextual information.**  Most of the SOTA forecasting models rely solely on historical data; these models can significantly benefit from our ContextFormer fine-tuning. The technique could be extended to adapt large time-series foundational models like TimesFM and Chronos on dataset-specific contextual information.
Currently, the most common strategy to adapt pre-trained models like TimesFM to contextual information is through fitting exogenous linear models onto the covariates, but these models are limited in their ability to capture complex correlations and also depend on the availability of future covariate values, making them unsuitable for many real-world problems.

2) **Ability to handle diverse metadata types and base architecture.** Our strategy can adapt and enhance any pre-existing forecasting model, regardless of the architecture, by utilizing any available contextual information—be it continuous or discrete, time-varying or time-invariant.

3) **Plug-and-play fine-tuning approach.** Our setup allows forecasters to incorporate domain-specific contextual information alongside the time-series features learned during pre-training, providing a guarantee for the context-aware model to be at least as good as the context-agnostic one without having to modify the entire model.

## **Revisions.**

1) **Related Works:** We have updated **Section 2 (Page 3)** to include details for the relevant literature on context-aware models, such as TFT, NBEATSx, and Timexer, which can utilize exogenous covariates.

1) **Theoretical Motivation**: We have streamlined **Section 5.1** to enhance clarity and conciseness while emphasizing the importance of theoretical analysis.

1) **Methodology**: We have updated the details on metadata embeddings to effectively clarify the processing methods. Additionally, we have added references to **Appendix D.2**, **Table 8**, and **Table 9**, pointing to the details of architectural implementations.

1) **Experiments:** We have included a comparison of our models against TiDE and TimeXer in **Table 2 (Page 8),** along with an additional dataset (**ETT**). **The ContextFormer-enhanced models outperform existing context-aware baselines like TimeXer & TiDE on the majority of the datasets. ContextFormer achieves up to 26% improvement over TimeXer (with an average of 6.6% across all settings) and up to 8.6% improvement over TiDE (averaging 4.1% across all settings) in terms of MSE.**  Both the baselines and the  ContextFormer-enhanced models take the same input (history + metadata).

1) **Appendix**: We have added comprehensive descriptions of **TiDE** and **TimeXer** in **Appendix D.2**, alongside a comparison with foundational models like **Chronos** and **TimesFM** in **Appendix E.2**. **Appendix C.6** now includes information about the newly included ETT dataset, while **Appendix C.8** has been updated to include details on metadata curation for the Bitcoin-News dataset, specifying the API name and the exact data period.

---

### Meta-Review · Area_Chair_tvSF · 2024-12-21

**Metareview:**

In this paper, the authors propose an approach to leveraging contextual features in time series prediction. Reviewers found the paper well-written but noted that its contribution is limited, the theoretical analysis lacks depth, and the experimental results are insufficient. They also commented that the work does not provide much insight into why the proposed approach effectively addresses the problem. Based on these concerns, I believe this work falls below the acceptance threshold.

**Additional Comments On Reviewer Discussion:**

In the rebuttal, the authors provided additional descriptions and details about the experiments. However, two reviewers maintained their view that the work is below the bar, and the other reviewers did not respond.

---

### Decision · Program_Chairs · 2025-01-22

Reject